# Empowering Active Learning for 3D Molecular Graphs with Geometric Graph Isomorphism

## Abstract

Molecular learning is pivotal in many real-world applications, such as drug discovery. Supervised learning requires heavy human annotation, which is particularly challenging for molecular data, *e.g.*, the commonly used density functional theory (DFT) is computationally very expensive. Active Learning (AL) automatically queries labels for most informative samples, thereby remarkably alleviating the annotation hurdle. In this paper, we present a novel and powerful AL paradigm for molecular learning, where we treat molecules as 3D molecular graphs. Specifically, we propose a new diversity sampling method to eliminate mutual redundancy built on distributions of 3D geometries. We first propose a set of new 3D graph isometrics for 3D graph isomorphism analysis. Our method is provably more powerful than the geometric Weisfeiler-Lehman (GWL) test. The moments of the distributions of the associated geometries are then extracted for efficient diversity computing. To ensure our AL paradigm selects samples with maximal uncertainties, we carefully design a Bayesian geometric graph neural network to compute uncertainties specifically for 3D molecular graphs. We pose active sampling as a quadratic programming (QP) problem using the novel components and conduct extensive experiments on the QM9 dataset. Results demonstrate the effectiveness of our AL paradigm, as well as the proposed diversity and uncertainty methods.

## 1 Introduction

Molecular representation learning is essential for a variety of real-world applications, such as molecular design, drug discovery, material design, *etc*. In recent studies, molecules have been formulated as 3D graphs, based on the evidence that 3D spatial information is crucial to determine the properties of molecules (Liu et al., 2019; Townshend et al., 2019; Axelrod & Gomez-Bombarelli, 2020). In a 3D graph, atoms are nodes, each of which is associated with the Cartesian coordinates in 3D space. A predefined cut-off distance can be used as a threshold to determine if there is an edge between two nodes in the 3D graph. With the advance of deep learning, 3D graph neural networks (GNNs) have been developed to learn from 3D molecular graph data (Schütt et al., 2017; Satorras et al., 2021; Gasteiger et al., 2020; Liu et al., 2021; Gasteiger et al., 2021; Thomas et al., 2018; Liao & Smidt, 2022). These models are data-hungry and necessitate a large amount of annotated training data to attain good performance. However, annotation usually consumes excessive manpower, which is particularly challenging for molecules, *e.g.*, the commonly used density functional theory (DFT) for molecular energy computing (Hohenberg & Kohn, 1964) is very expensive, inducing the complexity of $O(n_e^3)$, where $n_e$ is the number of electrons. As a concrete example, DFT can be hundreds of thousands of times slower than a reasonably good GNN for inference (Gilmer et al., 2017).

*Active Learning (AL)* algorithms automatically identify the salient and exemplar samples from large amounts of unlabeled data (Settles, 2009; Ren et al., 2021). This tremendously reduces the human annotation effort in inducing a deep neural network, as only the few samples that are identified by the algorithm need to be labeled manually. Further, since the deep network gets trained on the representative samples from the underlying data population, it typically depicts better generalization capability than a passive learner, where the training data is selected at random. Deep AL has been used with remarkable success in a variety of applications, such as computer vision (Yoo & Kweon, 2019; Sinha et al., 2019), natural language processing (Zhang et al., 2022), medical diagnosis (Blanch et al., 2017), and anomaly detection (Pimentel et al., 2020) among others. However, a principled AL algorithm for 3D molecular graph learning is currently lacking.

In this paper, we propose a powerful AL paradigm for 3D molecular graphs. We formulate a criterion based on uncertainty and diversity, which ensures that the queried molecules are the ones where the graph learning model has maximal uncertainty about the labels, and are also mutually diverse to avoid duplicate sample queries. In particular, diversity computing for 3D graphs is challenging and the *difficulties are twofold*. Firstly, the AL pipeline requires to compute the difference between any two 3D molecular graphs, which could have different planar molecules (entangling different atom numbers, *etc*) in most cases. Secondly, the 3D shape (geometry) of a 3D graph should be captured completely for expressive geometric representations for accurate diversity computing. To tackle these challenges, we propose a novel diversity sampling method for 3D molecular graphs built on distributions of important 3D geometries. We particularly propose a set of new 3D graph isometrics for 3D graph isomorphism. Our geometric modeling method is provably more powerful than the Geometric Weisfeiler-Leman (GWL) test (Joshi et al., 2023) in distinguishing and representing 3D graph geometries. This indicates our approach sets an upper bound to the expressive power of any existing equivariant geometric models. The moments of the distributions of the associated geometries (*e.g.*, reference distances, triangles) are extracted for accurate and efficient diversity computing. In addition, to ensure our AL paradigm selects samples with maximal uncertainties, we carefully design a Bayesian geometric graph neural network specifically for 3D graph uncertainty computing. Our method is shown to be effective and efficient based on a set of ground approximations. With our novel components, we pose the sample selection as a quadratic programming (QP) to identify exemplar molecules to be annotated. Our method is easy to implement and can be applied in conjunction with any 3D GNN architecture. We conduct extensive experiments on four properties of the QM9 dataset. Results demonstrate the effectiveness of our AL paradigm, as well as the proposed diversity and uncertainty methods.

## 2 METHODS

### 2.1 DIVERSITY COMPUTING FOR 3D MOLECULAR GRAPHS

In molecular AL tasks, diversity sampling is important for eliminating redundancy, thereby wisely leveraging the annotation budget. The model's capability of capturing the 3D shape diversity among molecules is crucial for informed sampling. A particular challenge lies in that, a diversity measure for two 3D molecules with different planar graphs is indispensable. Methods for diversity measures for 3D molecules with the same planar graph have been developed (Kumar & Zhang, 2018; Kearnes et al., 2016; Gfeller et al., 2013), but a diversity method for two 3D molecules with different planar graphs (entailing different atoms, *etc*) is demanding. Inspired by the USR method (Ballester & Richards, 2007), we propose a novel solution to achieve the goal from the distribution perspective. Generally, we develop a set of new *isometrics* for expressive representations of 3D molecular graphs, after which the distributions of geometries associated with the isometrics are obtained for diversity computing.

#### 2.1.1 ISOMETRICS OF 3D MOLECULAR GRAPHS

As the first step, we introduce a set of new *isometrics* as a basis, aiming at expressive representations of 3D graphs. As we focus on 3D geometry of molecules in this section, for simplicity, we use 3D point clouds to illustrate our ideas. Let $A = \{a_1, a_2, ..., a_n\}$ and $B = \{f(a_1), f(a_2), ..., f(a_n)\}$ be two sets representing 3D point clouds. Here, each $a_i$ in $A$ is associated with a positional vector $\vec{a_i} = (x_{a_i}, y_{a_i}, z_{a_i})$ in 3D space. $f$ denotes a bijective mapping between $A$ and $B$. Then, similarly, each point $f(a_i)$ in $B$ is associated with a positional vector $f(\vec{a_i}) = (x_{f(a_i)}, y_{f(a_i)}, z_{f(a_i)})$. Two 3D point clouds $A$ and $B$ are said to have isometric transformation when, given a group $SE(3)$, $\exists \gamma \in SE(3)$ such that $A = \gamma B$. We further choose or compute a reference point (*e.g.*, centroid) for each point cloud, denoted as $r_1$ and $r_2$, respectively. Without loss of generality, we use $a_{\text{far}}$ to denote the farthest point from the reference point in point cloud $A$. Below, we will define three levels of isometrics, each of which fulfills an isometrical mapping between $A$ and $B$. To satisfy *any* isometric, there needs to exist a bijective function $f : A \to B$, such that $h_{f(a)} = h_a$ for any node $a \in A$. Here, $h_{f(a)}$ and $h_a$ denote the node feature vectors for $f(a)$ and $a$, respectively.

**Reference Distance Isometric:** If there exists a collection of global group elements $\gamma_i \in SE(3)$, such that $(r_2, f(a_i)) = (\gamma_i r_1, \gamma_i a_i)$ for each point $a_i \in A$, $A$ is reference distance isometrical to $B$.

Reference distance isometric involves the Euclidean distance between any atom in the molecule and the predefined reference point.

**Triangular Isometric:** If there exists a collection of global group elements $\gamma_i \in SE(3)$, such that $(r_2, f(a_{\text{far}}), f(a_i)) = (\gamma_i r_1, \gamma_i a_{\text{far}}, \gamma_i a_i)$ for each point $a_i \in A$, $A$ is triangular isometrical to $B$.

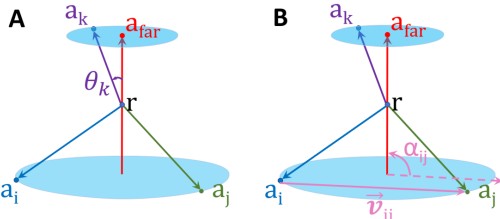

With reference point $r$, we define the reference vector $\vec{v}_0$ as $r$ pointing to the farthest point $a_{\text{far}}$ in a 3D molecule. Based on reference distance isometric, triangular isometric further involves the angle between $\vec{v}_0$ and any other vector pointing

Figure 1: The illustrations of encoding the molecular triangular and cross angular isometrics.

from $r$ to any other point in the molecule, computed as $\theta_k = \cos^{-1}\left(\frac{\vec{v}_0 \cdot \vec{v}_k}{\|\vec{v}_0\|\|\vec{v}_k\|}\right)$, where $\vec{v}_k$ denotes vectors originating from $r$ and directed towards $k^{\text{th}}$ atoms in the molecule. The process is illustrated in part A of Fig. 1. For a molecule with $N$ nodes, we compute $N-1$ angles. Essentially, such angles provide insights into the spatial arrangement of atoms with respect to the pre-assigned reference vector.

**Cross Angular Isometric:** If there exists a collection of global group elements $\gamma_{ij} \in SE(3)$, such that $(r_2, f(a_j), f(a_i)) = (\gamma_{ij} r_1, \gamma_{ij} a_j, \gamma_{ij} a_i)$ for all $a_i, a_j \in A$ $(i \neq j)$, $A$ is cross angle isometrical to $B$.

Beyond the angles in triangular isometric as well as based on reference distance isometric, cross angular isometric further considers angles formed by any two atoms in the molecule with respect to the reference vector as above. Specifically, for every pair of atoms $i$ and $j$, a vector $\vec{v}_{ij}$ is formed from $i$ to $j$. With the reference vector $\vec{v}_0$, the cross angle is computed as $\alpha_{ij} = \cos^{-1}\left(\frac{\vec{v}_0 \cdot \vec{v}_{ij}}{\|\vec{v}_0\|\|\vec{v}_{ij}\|}\right)$. This approach, as depicted in part B of Fig. 1, essentially reflects the torsion information (angles between two planes) in a molecular structure. For a molecule with $N$ nodes, we compute $N(N-1)/2$ cross angles with the complexity of $O(N^2)$. However, the commonly used way to compute torsion angles involves 3-hop neighborhood in a graph, inducing a much larger complexity of $O(N^3)$.

Notably, in this work, we propose the aforementioned three isometrics to define a complete isometry space, leading to the computing of the associated geometries (distances, angles) in Euclidean space for molecular diversity. There exist other studies to delineate isometry space, such as the first and second fundamental forms(Gallier, 2011). Compared with existing methods, the advantages of our solution are twofold. Firstly, our method represents a straightforward solution that can be naturally fitted into existing pipelines for molecular similarity computing. The commonly used methods, such as the Ultrafast Shape Recognition (USR) algorithm (Ballester & Richards, 2007) and Gaussian overlay-based methods (Rush et al., 2005; Hawkins et al., 2007), are all developed based on geometries (distances, angles) in Euclidean space. Secondly, our method is more efficient in capturing the essential geometric features. Taking the first and second fundamental forms as an example, these methods involve complex computations of a surface's differential properties, which could be expensive, especially for non-smooth surfaces that are common in molecular data. Specifically, estimating these forms accurately may lead to computational complexities up to $O(N^3)$, depending on the technique used for approximating surface derivatives and curvatures.

Next, we propose **Theorem 1** to indicate the relationship between these three isometrics as below.

**Theorem 1.** *If $A$ and $B$ are Triangular Isometric, then $A$ and $B$ are Reference Distance Isometric; If $A$ and $B$ are Cross Angular Isometric, then $A$ and $B$ are Triangular Isometric.*

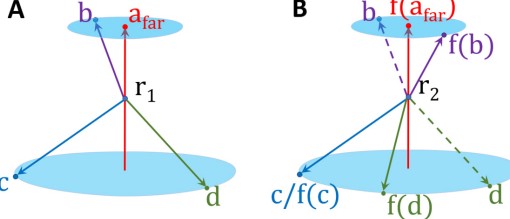

The proof of **Theorem** 1 can be found in Appendix A.2. Generally, we define three levels of isometrics for graph isomorphism. *Reference distance isometric* ensures that the Euclidean distance between each point and a predefined

Figure 2: $A$ and $B$ are triangular isomorphic but not cross angular isomorphic. The angles $\angle b r_1 a_{far}$, $\angle c r_1 a_{far}$, and $\angle d r_1 a_{far}$ in structure $A$ are equal to the angles $\angle f(b) r_2 f(a_{far})$, $\angle f(c) r_2 f(a_{far})$, and $\angle f(d) r_2 f(a_{far})$ in structure $B$, respectively. However, the cross angle $\angle d r_1 c$ is not equal to the cross angle $\angle f(d) r_2 f(c)$. reference point is consistent in two different point clouds.

*Triangular isometric* further manifests the spatial arrangement of atoms referring to the pre-assigned pivot. Built on *Triangular isometric*, *cross angular isometric* then reflects the spatial torsion angle information. An illustrative example for *triangular isometric* and *cross angular isometric* is also given in Fig. 2. Clearly, cross angular isometric represents the strictest isometry among the three. In the following Sec. 2.1.2, we show that a designed geometric representation based on *cross angular isometric* can exhibit great expressive power.

### 2.1.2 Expressive Power of Our Geometric Representations

In this section, we aim to formally elucidate the expressive power of a geometric representation based on our developed isometrics in Sec. 2.1.1. Generally, we use $GR$ to denote any geometric representation to capture molecular structures, and function $\zeta$ be the mapping function associated with $GR$. We explore the Geometric Weisfeiler-Leman (GWL) test (Joshi et al., 2023), and then leverage GWL to measure the expressiveness of a $GR$. GWL test is an extension of the classic WL Test, enhancing its capabilities by incorporating both the topological structure of the graph and the geometric attributes of its vertices. Such an integration allows the GWL test especially apt for evaluating all 3D graph representation methods. Similar to the regular WL test, GWL test imposes an upper bound to the expressive power of 3D GNNs, *i.e.*, if GWL test fails to distinguish two 3D graphs, then all existing 3D GNNs would also fail. See details of the GWL test in Appendix A.1.

Let the relation $A \not\cong B$ indicate that two structures are not *reference distance isometric* in terms of their 3D geometries. We propose **Theorem 2** as follows:

**Theorem 2.** *A geometric representation GR has the same expressive power as the Geometric Weisfeiler-Leman (GWL) Test in distinguishing non-isomorphic molecular structures if the associated mapping function $\zeta$ can map any two molecular structures $A$ and $B$ into two distinct representations (i.e., $\zeta(A) \neq \zeta(B)$) if and only if $A \not\cong B$.*

The proof of **Theorem** 2 is provided in Appendix A.2. Next, we use $GR_{\text{ours}}$ to denote the geometric representation based on the isometrics developed in Sec. 2.1.1, From **Theorem** 1, we can easily conclude that cross angular isometric is the strictest isometric among the three. Naturally, we formulate $GR_{\text{ours}}$ as a set containing all reference distances, triangles, and cross angles in a 3D graph. We show $GR_{\text{ours}}$ adheres to the constraints stipulated in **Theorem** 2 through the Proposition below.

**Proposition 1.** $GR_{ours}$ *can map any two different molecular structures $A$ and $B$ into two distinct representations (i.e., $\zeta(A) \neq \zeta(B)$) if and only if $A \not\cong B$.*

The proof of **Proposition** 1 is provided in Appendix A.2. Based on both **Theorem** 2 and **Proposition** 1, it is evident that our geometric representation $GR_{\text{ours}}$ possesses an expressiveness at least as powerful as the GWL test. Moreover, through the proposition below, we rigorously show $GR_{\text{ours}}$ possesses greater expressive power than the GWL test.

**Proposition 2.** $GR_{ours}$ *is strictly more expressive than the Geometric Weisfeiler-Leman (GWL) test in distinguishing non-isometric point clouds.*

The proof of **Proposition** 2 can be found in Appendix A.2. In conclusion, the molecular geometric representation method $GR_{\text{ours}}$ developed in this work has the greater expressive power than the Geometric Weisfeiler-Leman (GWL), which indicates our diversity sampling method is accurate enough to capture the 3D shape diversity among different molecules. Notably, as mentioned before, GWL test sets the upper bound to any existing 3D GNNs. Apparently, our geometric representation method $GR_{\text{ours}}$ is provably more expressively powerful than any existing 3D GNN for learning geometric representations. Essentially, the three isometrics associated with $GR_{\text{ours}}$ define expressiveness at different levels. For example, as only considering distance information, a well pretrained SchNet is upper bounded by reference distance isometric (but not triangular isometric or cross angular isometric); as a more powerful model than SchNet, a well pretrained DimeNet is upper bounded by triangular isometric (but not cross angular isometric). Additionally, accurate geometric representation learning requires a perfect pretrained 3D GNN model, which is also hard to guarantee in practice. Our isomorphy study provides a model-agnostic diversity computing solution, avoiding the need of a perfectly pretrained 3D GNN model, as well as achieving a theoretically guaranteed upper bound of the expressiveness of all existing 3D GNN models.

### 2.1.3 Final Distributional Representations

Based on the isomorphy study in Sec. 2.1.1, we obtain our geometric representation method $GR_{\text{ours}}$ and prove $GR_{\text{ours}}$ possesses greater expressive power than any existing 3D GNN models in Sec. 2.1.2. In this section, we aim to extract the distributions of the entangled three geometries in $GR_{\text{ours}}$,

including reference distances, triangles, and cross angles, for diversity computing. Fortunately, we have the theorem (Hall, 1983) implying that the sequence of translated moments can be used to determine the original distribution. Following the USR work (Ballester & Richards, 2007), for each of the three aforementioned geometries, we also use four reference points to reflect the "translated" geometries; those are, the centroid that is denoted as ctd and computed by the mean position of all the atoms in the 3D molecule, the point closest to the centroid (denoted as cst), the point farthest from the centroid (denoted as fct), and the point farthest from fct (denoted as ftf). For each reference point, we use a set of moments, including mean, variance, skewness, and kurtosis, which describe a distribution from different angles, *e.g.*, skewness indicates the asymmetry and kurtosis describes the tailedness of a distribution. Detailed formulae for these moments can be found in the Appendix A.3. Notably, we compute these translated moments for all three entangled geometries as above. Eventually, we obtain summarized representations of distributions over geometries of 3D graphs, capturing essential characteristics of a molecule's shape.

We use cross angles as an example to describe the final distributional vector. For a molecule with $N$ atoms, as shown in Fig. 1, we can obtain a set of cross angles $[\alpha_{ij}^{\text{ref}}]_{i\neq j,0<i,j<N}^{N(N-1)/2}$ for a reference point (*e.g.*, ctd). After applying statistical moments as an approximation, we can obtain a 4-dimensional vector $\overrightarrow{M_{\text{ref}}^{\text{ca}}} = [m_{\text{ref}}^{\text{ca}}, v_{\text{ref}}^{\text{ca}}, s_{\text{ref}}^{\text{ca}}, k_{\text{ref}}^{\text{ca}}]$, where the four elements denote the mean, variance, skewness, and kurtosis for this reference point, respectively. We perform a similar process for all four reference points mentioned above. By doing this, we can obtain four 4-dimensional vectors including $\overrightarrow{M_{\text{ctd}}^{\text{ca}}}$, $\overrightarrow{M_{\text{cst}}^{\text{ca}}}$, $\overrightarrow{M_{\text{fct}}^{\text{ca}}}$, and $\overrightarrow{M_{\text{ftf}}^{\text{ca}}}$, which are then concatenated together, resulting in the final 16-dimensional vector to represent the distribution of cross angles. We repeat the similar process for reference distances and triangles, and then all three corresponding 16-dimensional vectors are further concatenated as a 48-dimensional distributional vector to represent the geometric information of the input molecule. The 48-dimensional distributional vectors are then used to compute the diversity matrix as in Sec. 2.1.3. Finally, for any two molecules $n_1$ and $n_2$ in the dataset with $N$ molecules, we perform the inner product on their distributional vectors to achieve the similarity, and then use $1-$ similarity to obtain the final value $D_{n_1n_2}$ as the diversity measure between them. Finally, a matrix $D \in \Re^{N \times N}$ is obtained, which contains the diversity between every pair of molecules.

## 2.2 UNCERTAINTY COMPUTING FOR 3D MOLECULAR GRAPHS

In Sec. 2.1, we develop an effective method for diversity computing among different 3D molecular graphs. However, as mentioned in Sec. 1 and Sec. 3, in addition to selecting diverse molecules, it is important to select molecules where the model has maximal prediction uncertainty about the labels, so as to append maximal information to the model. To this end, we develop a principled pipeline to compute uncertainties for 3D molecular graphs in this section. Uncertainty qualification is well studied in planar graph analysis (Hirschfeld et al., 2020), but an effective and principled paradigm for 3D molecular graphs is currently lacking. Additionally, existing methods, such as Bayesian neural networks (BNNs) (Lampinen & Vehtari, 2001; Titterington, 2004; Goan & Fookes, 2020) and deep model ensemble methods (Lakshminarayanan et al., 2017; Huang et al., 2017), are excessively computationally expensive, limiting their capacity in 3D graph analyses. In this work, we develop an effective and efficient method, known as Bayesian geometric graph neural network (BGGNN), that takes a 3D graph as input and produces the demanding properties as well as uncertainty values, *e.g.*, mean and variance.

Formally, a 3D graph is represented as $\mathbf{G} = (V, E, P)$, where $V$ denotes the set of vertices (atoms), $E$ denotes for the set of edges (bonds), and $P$ stands for the set of Cartesian coordinates for each atoms. A 3D molecular graph is associated with a set of properties, denoted as $\mathbf{O}$. Recently, researchers have developed 3D GNNs, such as SchNet (Schütt et al., 2017), DimeNet (Gasteiger et al., 2020), SphereNet (Liu et al., 2021), and GemNet (Gasteiger et al., 2021), for 3D graph representation learning. The likelihood of a 3D GNN can be represented as $p_{\text{3DGNN}}(\mathbf{O} \mid \mathbf{G}, \mathbf{w})$, where 3DGNN indicates any existing 3D GNN and $\mathbf{w}$ denotes the set of parameters of the used 3D GNN. We also use $p_{\text{3DGNN}}(\mathbf{w})$ to represent the prior distribution for the parameters. Assume we collect a new input and output pair, denoted as $\mathbf{g}^*$ and $\mathbf{o}^*$. Then based on the conventional Bayesian theorem, Bayesian inference for this new output $\mathbf{o}^*$ is given by

$$p_{\text{3DGNN}}(\mathbf{o}^* \mid \mathbf{g}^*, \mathbf{G}, \mathbf{O}) = \int_{\mathbb{R}^n} p_{\text{3DGNN}}(\mathbf{o}^* \mid \mathbf{g}^*, \mathbf{w}) \, p_{\text{3DGNN}}(\mathbf{w} \mid \mathbf{G}, \mathbf{O}) d\mathbf{w}, \quad (1)$$

where $\mathbb{R}^n$ is the whole space of $n$ parameters in 3DGNN. It's infeasible to perform the above integration on $\mathbb{R}^n$ due to prohibitive computational cost. To tackle this, the variational inference

method is introduced to approximate $p_{\text{3DGNN}}(\mathbf{O} \mid \mathbf{G}, \mathbf{w})$ with the parameterized $q_\theta(\mathbf{w})$ through minimizing the Kullback-Leibler (KL) divergence between these two distributions. After applying Bayesian theorem once more, the minimization objective becomes

$$\mathcal{L}_{\text{VI}}(\theta) = -\int_{\mathbb{R}^n} q_\theta(\mathbf{w}) \log p_{\text{3DGNN}}(\mathbf{O} \mid \mathbf{G}, \mathbf{w})d\mathbf{w} + \text{KL}\left(q_\theta(\mathbf{w})\|p_{\text{3DGNN}}(\mathbf{w})\right). \quad (2)$$

To completely avoid the integration over the whole parameter space, the MC-dropout method (Gal & Ghahramani, 2016; Srivastava et al., 2014) is further used in our BGGNN. Specifically, it employes the Monte-Carlo estimator (Gal et al., 2016; Gal & Ghahramani, 2016) to approximate the integration by performing summation over the sampled models. In practice, researchers implement an MC-dropout network by using dropout as the network's regularization(Gal & Ghahramani, 2016). Following this, we propose to insert dropout layers after the linear layers in our used 3DGNN as an effective yet efficient estimation of Bayesian inference.

Now as we have obtained the variational predictive distribution of a new output with $q_\theta(\mathbf{w})$, we can easily compute the predictive mean and variance of this distribution. For the molecular property prediction tasks, after we sample $N$ outputs from the same input, the heteroscedastic predictive uncertainty is then given by

$$\widehat{\sigma^2}\left(\mathbf{o}^* \mid \mathbf{g}^*\right) = \frac{1}{N}\sum_{n=1}^{N}(\hat{\mathbf{o}}_n^*)^2 - \left(\frac{1}{N}\sum_{n=1}^{N}\hat{\mathbf{o}}_n^*\right)^2 + \frac{1}{N}\sum_{n=1}^{N}\widehat{\sigma}_n^2, \quad (3)$$

where $\hat{\mathbf{o}}_n^*$ is the $n^{th}$ sampled output and $\widehat{\sigma}_n^2$ is the variance that is the same among all the data samples. By doing this, we can obtain an uncertainty value (variance) for each molecule. Additionally, built on a 3D GNN, our BGGNN can faithfully produce a set of molecular properties $\mathbf{O}$.

Practically, any of the existing 3D GNN can be used as the backbone network for property prediction and uncertainty computing. In this study, we employ SphereNet (Liu et al., 2021) as our 3DGNN, owing to its completeness in incorporating 3D geometric information. We apply dropout layers onto the linear layers of SphereNet for Bayesian inference in our BGGNN. To allow more accurate AL selections, we particularly employ the concrete dropout with a learnable dropout rate (Gal et al., 2017) in our BGGNN. Overall, our method is shown to be an effective and efficient paradigm for 3D graph uncertainty computing, as further empirically demonstrated in Sec. 4.

### 2.3 ACTIVE SAMPLING

A schematic diagram of our active sampling framework is depicted in Fig. 6 and described in A.4 in Appendix. Specifically, in Sec. 2.1, we obtain the matrix $D \in \Re^{N \times N}$ containing the mutual diversity between every pair of unlabeled molecules, where $N$ is the number of unlabeled molecules. In Sec. 2.2, we employ our designed BGGNN to achieve the vector $r \in \Re^{N \times 1}$ quantifying the prediction uncertainty score of each unlabeled molecule. In the AL setting, our objective is to select a batch of $k$ unlabeled molecules (where $k$

$$\max_z \quad z^\top r + \lambda z^\top D z$$
$$s.t. \quad \sum_{i=1}^{N} z_i = k$$
$$z_i \in \{0, 1\}, \forall i, \quad (4)$$

is the pre-determined query batch size) with high prediction uncertainty and high mutual diversity among them. Let $z \in \{0, 1\}^{N \times 1}$ be a binary vector with $N$ entries which denotes whether the unlabeled molecule $x_i$ will be included in the batch ($z_i = 1$) or not ($z_i = 0$). The molecule selection can thus be posed as the following optimization problem as in Eq. 4, where $\lambda$ is a weight parameter governing the relative importance of the two terms. This is a standard quadratic programming (QP) problem; we relax the integer constraints into continuous constraints and solve the problem using an off-the-shelf QP solver. In this work, we employ the widely used Operator Splitting Quadratic Program (OSQP) (Stellato et al., 2020) to solve the QP problem in Eq. 4. We then apply a greedy approach to project the continuous solution back to the binary space, where the $k$ highest entries of the continuous solution vector are set to 1 and the remaining to 0. Such an approach is commonly used to convert continuous solutions obtained from a QP solver to binary solutions in AL Chattopadhyay et al. (2013); Wang & Ye (2013). Notably, the predictions in main tasks (*e.g.*, molecular properties) are produced by our BGGNN built on SphereNet as in Sec. 2.2.

## 3 RELATED WORK

### 3.1 ACTIVE LEARNING

Active Learning (AL) is a well-researched problem in the machine learning community (Settles, 2009). There exist two commonly used strategies for AL sampling. Uncertainty based sampling queries unlabeled samples with the highest prediction uncertainties for annotation. Diversity/representativeness

based sampling aims to select the subset that can well represent the entire data distribution. A full review of the two AL sampling methods is provided in Appendix A.5.

## 3.2 MOLECULAR SHAPE SIMILARITY

Molecular shape similarity plays a pivotal role in drug discovery and virtual screening of compounds (Kumar & Zhang, 2018; Murgueitio et al., 2012; Shang et al., 2017). Methods predominantly fall into several categories (Kumar & Zhang, 2018), including descriptor-based methods (Schreyer & Blundell, 2012; Cannon et al., 2008; Li et al., 2016; Armstrong et al., 2009; Zhou et al., 2010), atom-centered gaussian-based methods (Haque & Pande, 2010; de Lima & Nascimento, 2013; Yan et al., 2013), surface-based methods (Hofbauer et al., 2004; Mavridis et al., 2007; Cai et al., 2012; Karaboga et al., 2013; Venkatraman et al., 2009; Sael et al., 2008), *etc*. Descriptor-based methods are notably represented by the Ultrafast Shape Recognition (USR) algorithm (Ballester & Richards, 2007), which uses statistical moments of the distance distribution to characterize molecular shapes. Gaussian overlay-based methods, with ROCS (Rush et al., 2005; Hawkins et al., 2007) being the most commonly used one, evaluate the maximum volume overlap between two molecules. Surface-based methods typically employ shape signatures (Zauhar et al., 2013) or shape histograms to delineate molecular surfaces for shape similarity assessment. Despite the progress, a principled and theoretically ground similarity method for 3D molecular graphs is currently lacking.

## 4 EXPERIMENTS

### 4.1 EXPERIMENTAL SETUP

**Data and Active Learning Setup**: We perform experiments on four properties of the QM9 (Ramakrishnan et al., 2014) benchmark dataset: *mu, alpha, homo, and lumo*. These properties have continuous values, making the prediction problem a regression task. We randomly divide the training set of $110,000$ molecules into three splits of size $25,000$ each. From each split, we randomly select $5,000$ molecules as the initial labeled set and the remaining $20,000$ molecules as the unlabeled set. In each AL iteration, we query $1,500$ molecules from the unlabeled set, which are labeled and appended to the labeled set. The model's performance is evaluated on a held-out validation set containing $10,000$ molecules. We save the best-performing model on the validation set and report its performance on the test set containing $10,831$ molecules. The process is repeated for 7 AL iterations, which is taken as the stopping criterion. The final results are averaged over the three splits to rule out the effects of randomness. The value of $\lambda$ in Eq. 4 is taken as 1, for the QP problem. The Mean Absolute Error (MAE) is used as the evaluation metric in this work.

**Implementation Details**: We use SphereNet (Liu et al., 2021) as the backbone model of our BGGNN in all the experiments. As mentioned in Liu et al. (2021), we set the optimal network configurations for the SphereNet and train the network for 150 epochs. We use the *Adam Optimizer* with the initial learning rate $5 \times 10^{-4}$ and scale it by a factor of 0.5 every 15 epochs. PyTorch is used for implementation, and models are trained using NVIDIA RTX A4500 20GB GPUs.

**Comparison Baselines**: We use four AL methods as baselines: *Random Sampling*, *Coreset* (Sener & Savarese, 2018), *Learning Loss* (Yoo & Kweon, 2019), and *Evidential Uncertainty* (Beluch et al., 2018; Amini et al., 2020). *Random Sampling* is

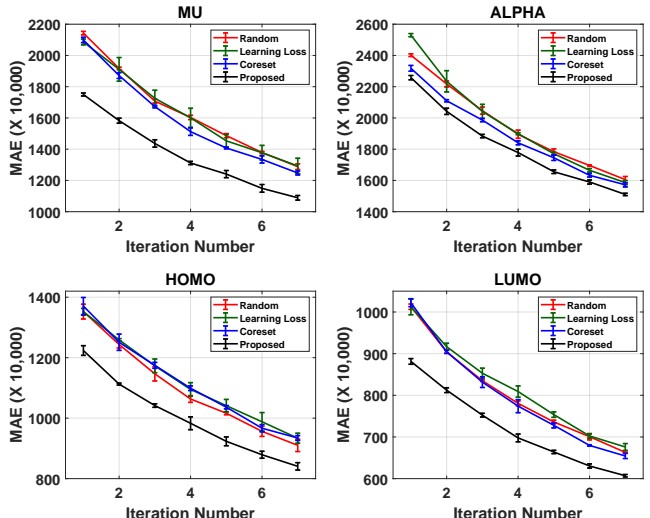

Figure 3: Active learning performance results. The graphs show the mean (averaged over 3 runs) and the errorbars for all the methods. Best viewed in color.

the default comparison baseline in AL research. *Coreset* and *Learning Loss* are two extensively used deep active learning algorithms for regression applications. *Evidential Uncertainty* is also a commonly used technique to quantify uncertainty for molecular property prediction, and was hence included as a comparison baseline.

## 4.2 ACTIVE LEARNING PERFORMANCE

The active learning performance results are depicted in Fig. 3. In each graph, the $x$-axis denotes the iteration number and the $y$-axis denotes the MAE on the test set. Our analysis revealed that *Evidential Uncertainty* depicted the worst performance and furnished significantly high error values for all the four properties, which obscured the difference in performance among the other methods in the plots. For better interpretation and understanding, we exclude the *Evidential Uncertainty* method from the plots here and present the results with this baseline in Sec. A.6 of the Appendix. The other baseline methods depict more or less similar performance, with *Coreset* marginally outperforming the other baselines. Our method comprehensively outperforms all the baselines. At any given AL iteration, it consistently attains a lower MAE compared to all the baselines. It also attains the least MAE after all the AL iterations, for all the four properties studied.

We also conducted statistical tests of significance using paired t-test to assess whether the improvement in performance achieved by our method is statistically significant. For this purpose, we compared the average MAE achieved by our method against each of the baselines individually. The results are reported in Table 1; each entry in the table denotes the p-value of the paired t-test between our method

Table 1: The table shows the p-values obtained using paired t-test between the results our method against each of the baselines for all the properties studied. *Here, L. Loss refers to Learning Loss.*

| Properties | Baselines | | | |
|---|---|---|---|---|
| | Random | L. Loss | Coreset | Evidential |
| *mu* | $7.54\times10^{-6}$ | $5.09\times10^{-5}$ | $1.51\times10^{-4}$ | $2.19\times10^{-7}$ |
| *alpha* | $1.06\times10^{-5}$ | $8.14\times10^{-4}$ | $4.27\times10^{-5}$ | $2.72\times10^{-4}$ |
| *homo* | $2.26\times10^{-5}$ | $8.36\times10^{-7}$ | $4.23\times10^{-6}$ | $1.71\times10^{-8}$ |
| *lumo* | $4.48\times10^{-5}$ | $1.25\times10^{-5}$ | $3.12\times10^{-4}$ | $2.39\times10^{-6}$ |

and the corresponding baseline (denoted in the columns) for the property studied (denoted in the rows). From the table, we note that the improvement in performance achieved by our method is statistically significant ($p < 0.05$) compared to all the baselines, consistently for all the four properties studied. These results unanimously corroborate the promise and potential of the proposed active sampling method to tremendously reduce the annotation cost in inducing a robust 3D graph neural network for molecular property prediction.

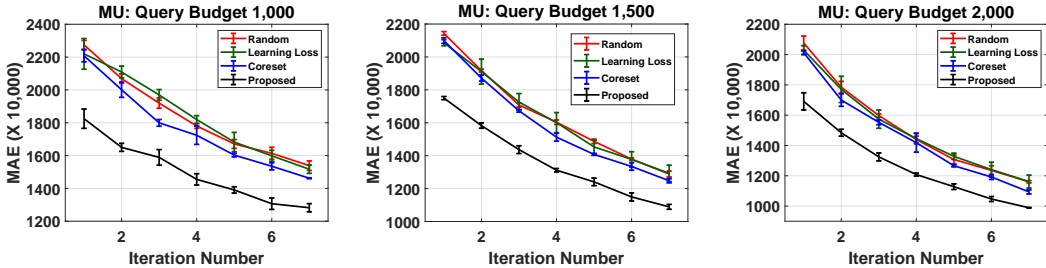

Figure 4: Study of query budget on the active learning performance. The graphs show the mean (averaged over 3 runs) and the errorbars for all the methods. The results with budget 1500 are the same as the those presented in Figure 3 and are included here for comparison. Best viewed in color.

## 4.3 STUDY OF QUERY BUDGET

The goal of this experiment is to study the effect of query budget (batch size) on the AL performance. The results on the **mu** property for query budgets $1,000$, $1,500$ and $2,000$ are depicted in Fig. 4. Since *Evidential Uncertainty* depicted much worse performance than all the methods, it was excluded

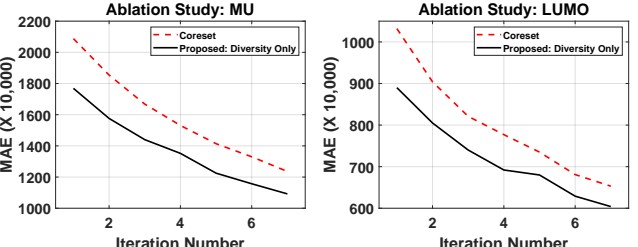

Figure 5: Ablation study results on the **mu** and **lumo** properties. Best viewed in color.

from this comparison. Our framework once again outperforms all the baselines consistently for all the query budgets and attains the least MAE after all the AL iterations. As before, we conducted a paired t-test to estimate the statistical significance of the obtained performance improvement. The results are presented in Sec. A.7 of the Appendix. From the p-values, we conclude that the error values furnished by our method are statistically significantly better ($p < 0.05$) than all the baselines, consistently for all the query budgets.

This shows the robustness of our framework to the query budget. These results are particularly significant from a practical standpoint as the available query budget in a real-world application is dependent on time, resources, and other constraints of the application.

### 4.4 ABLATION STUDIES

We conduct ablation studies to examine the power of our diversity computing method, as it is our primary contribution in this research. We perform experiments on the **mu** and **lumo** properties from two aspects. Firstly, we compare our framework with only the diversity term in Eq. 4 against *Coreset*, the state-of-the-art diversity-based AL technique. The results are reported in Fig. 5, from which we note that the diversity component of our framework consistently furnishes much lower MAE values than *Coreset* over all the AL iterations, for both properties. Secondly, we also conducted experiments where we compared the performance of our overall framework (using both uncertainty and diversity) against the baseline where only the uncertainty term in Eq. (4) was used for active sampling. The results revealed that removing the diversity term adversely affected the performance of our framework. A paired t-test revealed that the improvement in performance achieved by our diversity component is statistically significant ($p < 0.05$) for both these properties ($p = 0.0001$ for **mu** and $p = 0.04$ for **lumo**). These results show the usefulness of the proposed diversity metric in developing an AL framework to train a 3D GNN for molecular property prediction.

### 4.5 COMPUTATION TIME ANALYSIS

In this experiment, we analyze the computation time of all the methods studied in this paper. The average time taken to query a batch of unlabeled samples and update the SphereNet model (one active learning iteration) are shown in Table 2. For fair comparison, all the methods were run on the same NVIDIA RTX A4500 20GB GPU. We note that *Random, Learning Loss* and *Evidential Uncertainty* all have similar computation time. *Coreset* depicts the highest computation time, as it needs to solve a mixed integer programming (MIP) problem. The computation time of our framework is marginally less than *Coreset*, and approximately double that of the other three. Our analysis revealed that solving the QP problem is the main bottleneck of our method. Hence, we explored a strategy to improve the computation overhead of solving the QP problem by running it in the GPU (instead of the CPU) using the parallel implementation of the alternating direction method of multipliers, as detailed in Schubiger et al. (2020). This results in a substantial reduction of the computation time, as depicted in Table 2 (Ours (Fast QP)). The AL performance using the fast QP solver is very similar to that obtained using the original solver, and is presented in Fig. 8 in Sec. A.8 of the Appendix. Apparently, the performance studies in both Sec. 4.2 and Sec. A.8 show that our framework is much more accurate than these baselines. Given the large margin of performance improvement, we think the efficiency of our methods is acceptable. As solving QP problems is not the main technical novelty of this work, we will continue exploring faster QP solvers from literature for engineering purposes as future work.

Table 2: Average ($\pm$ std) time taken by each method for sample selection and training the SphereNet model (one iteration of AL). *Here, L. Loss refers to Learning Loss.*

| Selection Methods | | | | | |
|---|---|---|---|---|---|
| Random | L. Loss | Coreset | Evidential | Ours | Ours (Fast QP) |
| 53$\pm$4.5min | 56$\pm$2.1min | 2hr 7$\pm$3.5min | 56$\pm$2.3min | 1hr 58$\pm$3.7 min | 1hr 28$\pm$7.5 min |

## 5 CONCLUSION AND FUTURE WORK

In this paper, we present a novel active learning framework with the goal of reducing the annotation cost for learning from 3D molecules represented as 3D graphs. The sample selection is posed as a QP problem, which selects samples with high mutual diversity and high uncertainty. Novel diversity and uncertainty components are proposed specifically for 3D graphs, with strong empirical results demonstrating the promise and potential of our AL framework. As part of future work, we plan to apply our methods to molecular analysis where much more accurate but expensive annotation is required, such as computing ground states of molecular systems using the Schrödinger equation. DFT calculations are widely used in many practical applications but still involve approximations, as Schrödinger equation is prohibitively expensive and its use is limited in very small molecules. Our AL pipeline is anticipated to unleash greater potential in such extreme-scale applications. Additionally, given AL needs several interactions with each requiring the model is well trained, we test our methods on the commonly used but medium-scale QM9 dataset in this work. Even though we think the empirical studies are sufficient to support our theory, we still plan to test the scalability of our methods on large-scale molecule datasets, such as OC20 (Chanussot et al., 2021), in the future.

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

# A APPENDIX

## A.1 GEOMETRIC WEISFEILER-LEMAN (GWL) TEST

For the Geometric Weisfeiler-Leman (GWL) test, consider a graph $\mathcal{G}$ with its set of vertices represented as $\mathcal{V}(\mathcal{G})$ and its set of edges as $\mathcal{E}(\mathcal{G})$. A vertex in graph $\mathcal{G}$ is denoted by $i$, and $\mathcal{N}_i$ signifies the set of vertices adjacent to $i$. The color of vertex $i$ at iteration $t$ is given by $c_i^{(t)}$, and the geometric object for vertex $i$ at iteration $t$ is represented by $\boldsymbol{g}_i^{(t)}$.

The procedure for the GWL test is as follows:

1. **Initialization**: Each vertex $i$ is assigned an initial color $c_i^{(0)}$ and a geometric object $\boldsymbol{g}_i^{(0)}$, typically based on its local property or geometric attributes.

2. **Iterative Aggregation**: For each iteration $t \geq 1$, the geometric object of each vertex $i$ is updated to aggregate geometric information from its $t$-hop neighborhood, represented as $\boldsymbol{g}_i^{(t)}$, which includes the colors and geometric objects from the previous iteration of vertex $i$ and its neighbors.

3. **Color Update**: The color of each vertex $i$ at iteration $t$ is computed by aggregating the geometric information around vertex $i$ using a $\mathfrak{G}$-orbit injective and $\mathfrak{G}$-invariant function, denoted by I-HASH, i.e., $c_i^{(t)} := \mathrm{I}^{-\mathrm{HASH}^{(t)}}\left(\boldsymbol{g}_i^{(t)}\right)$.

4. **Termination**: The procedure terminates when colors do not change from the previous iteration or a predetermined maximum number of iterations is reached.

5. **Graph Comparison**: Finally, two geometric graphs $\mathcal{G}$ and $\mathcal{H}$ are geometrically non-isomorphic if there exists some iteration $t$ for which the sets of colors of their vertices are not equal, i.e., $\left\{\left\{c_i^{(t)} \mid i \in \mathcal{V}(\mathcal{G})\right\}\right\} \neq \left\{\left\{c_i^{(t)} \mid i \in \mathcal{V}(\mathcal{H})\right\}\right\}$.

## A.2 PROOFS OF THE THEOREMS AND PROPOSITIONS

**Theorem 1.** *If $A$ and $B$ are Triangular Isometric, then $A$ and $B$ are Reference Distance Isometric; If $A$ and $B$ are Cross Angular Isometric, then $A$ and $B$ are Triangular Isometric.*

*Proof.* **1. If $A$ and $B$ are Triangular Isometric**

Assume that $A$ and $B$ are Triangular Isometric, i.e., there exists a collection of global group elements $\gamma_i \in SE(3)$ such that

$$(r_2, f(a_{\mathrm{far}}), f(a_i)) = (\gamma_i r_1, \gamma_i a_{\mathrm{far}}, \gamma_i a_i)$$

for each point $a_i \in A$.

From the above assumption, it is clear that for each point $a_i \in A$, there exists a corresponding $\gamma_i \in SE(3)$ such that

$$(r_2, f(a_i)) = (\gamma_i r_1, \gamma_i a_i)$$

which is the condition for $A$ and $B$ to be Reference Distance Isometric. Hence, the statement is proved.

**2. If $A$ and $B$ are Cross Angular Isometric**

Assume $A$ and $B$ are Cross Angular Isometric, which means for all $a_i, a_j \in A$ with $i \neq j$, there exists a collection of global group elements $\gamma_{ij} \in SE(3)$ such that

$$(r_2, f(a_j), f(a_i)) = (\gamma_{ij} r_1, \gamma_{ij} a_j, \gamma_{ij} a_i).$$

From this assumption, for any point $a_i \in A$, there exists a corresponding $\gamma_i \in SE(3)$ (which is one of the $\gamma_{ij}$ where $j$ corresponds to the farthest point $a_{\mathrm{far}}$) such that

$$(r_2, f(a_{\mathrm{far}}), f(a_i)) = (\gamma_i r_1, \gamma_i a_{\mathrm{far}}, \gamma_i a_i).$$

This is precisely the condition for $A$ and $B$ to be Triangular Isometric. Hence, if $A$ and $B$ are Cross Angular Isometric, then $A$ and $B$ are also Triangular Isometric.

$\square$

**Theorem 2.** *A geometric representation $GR$ has the same expressive power as the Geometric Weisfeiler-Leman (GWL) Test in distinguishing non-isomorphic molecular structures if the associated mapping function $\zeta$ can map any two molecular structures $A$ and $B$ into two distinct representations (i.e., $\zeta(A) \neq \zeta(B)$) if and only if $A \not\cong B$.*

*Proof.* **1. GWL Distinguishable implies $GR$ Distinguishable:** Suppose two molecular structures, $A$ and $B$, are distinguishable by the GWL Test after k iterations. That is to say, the multiset of node colors remains the same through iteration 0 to $k - 1$ but becomes different in the $k^{th}$ iteration. According to the GWL Test in the Graph Compasison step, there exist $a_0 \in A$ and $f(a_0) \in B$ such that $\left\{\left\{ c_{a_0}^{(t)} \mid i \in A \right\}\right\} \neq \left\{\left\{ c_{f(a_0)}^{(t)} \mid f(a_0) \in B \right\}\right\}$. Then by the color update rule of GWL Test, this means that $a_0 \in A$ and $f(a_0) \in B$ have different distances from their respective reference points and thus $A$ and $B$ are non-Reference Distance Isometric. According to the constraints on $GR$ in the theorem, $\zeta(A)$ and $\zeta(B)$ are different and distinguishable.

**2. $GR$ Distinguishable implies GWL Distinguishable:** Suppose that, at the $k^{th}$ iteration, two molecular structures $A$ and $B$ are distinguishable by the representation method $GR$. This implies that the mappings $\zeta(A)$ and $\zeta(B)$ are different by being non-Reference Distance Isometric according to the constraints on $GR$ in the theorem. So we are able to find at least $a_0 \in A$, such that $(r_2, f(a_0)) \neq (\gamma r_1, \gamma a_0)$ holds for $\forall \gamma \in SE(3)$ which can be colored differently by the GWL Test when $g_i^{(t)}$ encodes geometrical information of its neighbors. So GWL Test is also able to distinguish between $A$ and $B$.

$\square$

**Proposition 1.** *$GR_{ours}$ can map any two different molecular structures $A$ and $B$ into two distinct representations (i.e., $\zeta(A) \neq \zeta(B)$) if and only if $A \not\cong B$.*

*Proof.* **1. Non Reference Distance Isometric implies $GR_{ours}$ Distinguishable:** If $A$ and $B$ are non-Reference Distance Isometric, then by definition we know that $\exists a_0 \in A$, such that $(r_2, f(a_0)) \neq (\gamma r_1, \gamma a_0)$ holds for $\forall \gamma \in SE(3)$. So $\zeta(A)$ and $\zeta(B)$ would give different representation as a result of different distance which implies $A$ and $B$ are $R$ distinguishable.

**2. $GR_{ours}$ Distinguishable implies Non Reference Distance Isometric:** If $A$ and $B$ are $R$ distinguishable, then $\zeta(A) \neq \zeta(B)$, which means there is difference in representation. This discrepancy arises from either Reference Distance, Triangle or Cross Angle. Then by different choices of reference points, we know that $A$ and $B$ are Non Reference Distance Isometric.

$\square$

**Proposition 2.** *$GR_{ours}$ is strictly more expressive than the Geometric Weisfeiler-Leman (GWL) test in distinguishing non-isometric point clouds.*

*Proof.* First we prove that $GR_{ours}$ satisfies the two constraints as below.

1. $GR_{ours}$ can distinguish any two point clouds $A$ and $B$ by cross angles; that is, $\zeta(A) \neq \zeta(B)$, if $A$ and $B$ are reference distance isometric but are non cross angular isometric.

2. The mapping function $\zeta$ associated with $GR_{ours}$ is injective.

Then we would prove that with the constraints satisfied, $GR_{ours}$ would be more expressive than GWL Test.

**1. $GR_{ours}$ satisfies the two constraints:**

1. $A$ and $B$ are reference distance isometric implies that there exists a collection of global group elements $\gamma_i \in SE(3)$ such that $(r_2, f(a_i)) = (\gamma_i r_1, \gamma_i a_i)$ for each point $a_i \in A$. Meanwhile, $A$ and $B$ are non cross angular isometric implies that $\exists a_i, a_j \in A (i \neq j)$, such

that $(r_2, f(a_j), f(a_i)) \neq (\gamma r_1, \gamma a_j, \gamma a_i)$ holds for $\forall \gamma \in SE(3)$. Given $GR_{\text{ours}}$ introduced in Sec. 2.1.1, this would result in difference in cross angles and thus in results of $GR_{\text{ours}}$; that is, $\zeta(A) \neq \zeta(B)$.

2. Consider two distinct 3D molecular graphs represented by point clouds $A$ and $B$. Since $A$ and $B$ are distinct, there exists at least one isometric, as defined in Section 2.1.1, that is violated by $A$ and $B$. In $GR_{\text{ours}}$, the distributions of geometries associated with the isometrics would then be different. That is to say, $\zeta(A) \neq \zeta(B)$. Therefore, the mapping function $\zeta$ associated with $GR_{\text{ours}}$ is injective, mapping distinct 3D molecular point cloud to distinct representations.

**2. $GR_{\text{ours}}$ is more expressive than GWL Test:** $GR_{\text{ours}}$ that satisfies the two constraints can distinguish any two different point clouds $A$ and $B$ in $R^{3n}$. Because for the two different point clouds $A$ and $B$, they must be non cross angular isometric and thus are distinguishable by the injective mapping $\zeta$. We now prove that $GR_{\text{ours}}$ is more expressive than GWL Test by showing that there are circumstances when the GWL Test is unable to distinguish between two distinct point clouds. We present three cases for which GWL Test would fail.

1. For a given iteration $k$ in the GWL Test, the primary cause of failure arises when, despite the point clouds not being entirely isometric, there exist local structures or geometrical features that are sufficiently similar. In such case, they might be assigned the same node color partition even after the $k^{th}$ iteration.

2. When the color update scheme in GWL Test depends only on local area scalar features which are the same between two point clouds, the test will thus assign the two point clouds the same node color partition.

3. For GWL Test without a pre-assigned iteration number, it may also fail as a result of Hash collision. Different inputs may be mapped to the same Hash value and thus result in the same node color partition between different point clouds.

These are the cases that GWL Test may fail to distinguish two distinct point clouds. Since there exist point clouds that GWL Test cannot distinguish while $GR_{\text{ours}}$ can, we can draw the conclusion that $GR_{\text{ours}}$ is more expressive than the GWL Test. $\square$

### A.3 STATISTICAL MOMENTS

The equations that we used for calculating four moments are as follows.

The **mean**, often referred to as the average, represents the sum of all data points divided by the number of data points and is given by

$$\text{Mean} = \frac{\sum_{i=1}^{n} x_i}{n}. \tag{5}$$

**Variance** measures the spread or dispersion of a dataset and is defined as

$$\text{Variance} = \frac{\sum_{i=1}^{n}(x_i - \text{Mean})^2}{n-1}. \tag{6}$$

**Skewness** gauges the asymmetry of a dataset's distribution. Here we sightly change its definition to be positive for convenience as

$$\text{Skewness} = \frac{\sum_{i=1}^{n}|x_i - \text{Mean}|^3/n}{\left\{\sum_{i=1}^{n}(x_i - \text{Mean})^2/(n-1)\right\}^{3/2}}. \tag{7}$$

**Kurtosis** assesses the "tailedness" of a dataset's distribution as

$$\text{Kurtosis} = \frac{\sum_{i=1}^{n}(x_i - \text{Mean})^4/n}{\left\{\sum_{i=1}^{n}(x_i - \text{Mean})^2/(n-1)\right\}^{2}}. \tag{8}$$

### A.4 Schematic Diagram of our Framework

A schematic diagram of our active sampling framework is depicted in Fig. 6. We are given a labeled training set $L$, an unlabeled set $U$ and a query budget $k$ for each active learning iteration. The SphereNet model is first trained on the labeled set $L$. In the second step, the trained model is applied on the unlabeled set to compute a prediction uncertainty of each unlabeled molecule, which is used to populate the uncertainty vector $r$; the diversity matrix $D$ is also computed in this step where $D(i, j)$ is the diversity between unlabeled molecules $x_i$ and $x_j$. Next, the QP problem is solved to select $k$ unlabeled molecules for annotation. These molecules are removed from the unlabeled set $U$ and appended to the labeled set $L$. The active sampling process is continued iteratively until some stopping criterion is satisfied (taken as 7 iterations in our work).

Note that, computing the diversity matrix $D$ in Step 3 needs to be executed just once for the whole process. Once we have the initial $D$, as more and more samples are queried through AL, we keep deleting the corresponding rows and columns from $D$ to derive the updated matrix.

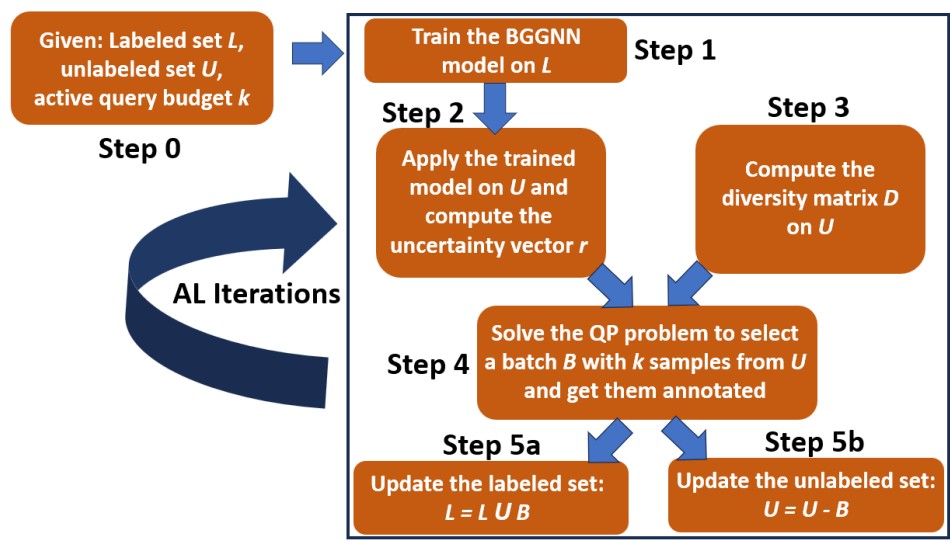

Figure 6: Schematic diagram of the proposed active learning framework.

### A.5 Related Work for Active Learning

Active Learning (AL) is a well-researched problem in the machine learning community (Settles, 2009). Uncertainty sampling is an important strategy for AL, where unlabeled samples with the highest prediction uncertainties are queried for annotation. Several techniques have been explored to compute the uncertainty, such as Shannon's entropy (Guo & Schuurmans, 2007; Li & Guo, 2013), the distance of a sample from the separating hyperplane for SVM classifiers (Tong & Koller, 2001), the disagreement among a committee of classifiers regarding the label of a sample (Freund et al., 1997; Gilad-Bachrach et al., 2005), among others (Hoi et al., 2006; HOI et al., 2008; Guo & Greiner, 2007; Freytag et al., 2014). With the advent of deep learning, Deep AL has attracted significant research attention (Hino, 2020; Ren et al., 2021), Entropy-based methods are developed as well (Wang & Shang, 2014; Ranganathan et al., 2017). Yoo & Kweon (2019) cascaded a task-agnostic loss learning module that queries samples with the highest predicted loss values. Huang et al. (2021) proposed a strategy based on temporal output discrepancy. Techniques based on adversarial training have also been explored (Sinha et al., 2019; Mayer & Timofte, 2020; Zhang et al., 2020; Zhu & Bento, 2017). Bayesian neural networks (BNNs) (Lampinen & Vehtari, 2001; Titterington, 2004; Goan & Fookes, 2020) and deep model ensemble (Lakshminarayanan et al., 2017; Huang et al., 2017) generally achieve superior performance but may induce excessive computational cost.

Diversity/representativeness based AL sampling has also been exploited. A core-set sampling technique proposed by Sener & Savarese (2017) queries a batch of samples such that a model trained on the queried subset is competitive for the remaining data samples. Diversity sampling has also been exploited in the context of Bayesian neural networks (Kirsch et al., 2019). Buchert et al. (2023) uses

diversity sampling, together with self-supervised representation learning to select an informative seed set for AL. Combinations of uncertainty/diversity/representativeness-based criteria have also been used as query functions in AL research (Chakraborty et al., 2015; Wu et al., 2022; Ash et al., 2020).

## A.6 RESULTS WITH THE EVIDENTIAL UNCERTAINTY BASELINE

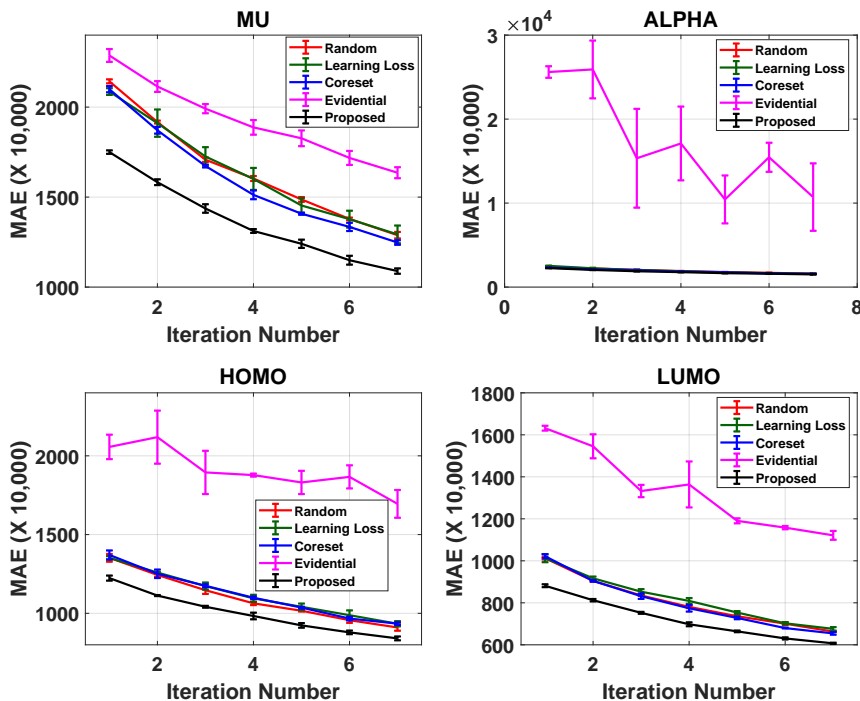

Figure 7: Active learning performance results. The graphs show the mean (averaged over 3 runs) and the errorbars for all the methods. Best viewed in color.

The active learning performance results on the four properties studied (*mu, alpha, homo, and lumo*) are depicted in Fig. 7. As mentioned in Sec. 4.2, we note that *Evidential Uncertainty* depicts significantly high error values than the other methods, for all the four properties.

## A.7 STATISTICAL TESTS OF SIGNIFICANCE FOR THE QUERY BUDGET EXPERIMENT

Table 3: The table shows the p-values obtained using paired t-test between the results our method against each of the baselines for the *mu* property for query budgets $1,000$, $1,500$ and $2,000$.

| Budget | Baselines | | | |
|---|---|---|---|---|
| | Random | Learning Loss | Coreset | Evidential |
| *1000* | $7.58 \times 10^{-6}$ | $1.05 \times 10^{-5}$ | $5.32 \times 10^{-5}$ | $2.46 \times 10^{-10}$ |
| *1500* | $7.54 \times 10^{-6}$ | $5.09 \times 10^{-5}$ | $1.51 \times 10^{-4}$ | $2.19 \times 10^{-7}$ |
| *2000* | $7.90 \times 10^{-5}$ | $1.74 \times 10^{-5}$ | $1.94 \times 10^{-4}$ | $1.77 \times 10^{-8}$ |

Table 3 reports the results of the statistical tests of significance for the study of query budget (presented in Sec. 4.3). Each entry in the table denotes the p-value of the paired t-test between our method and the corresponding baseline (denoted in the columns) for the query budget (denoted in the rows) for the *mu* property. From the table, we note that the improvement in performance achieved by our method is statistically significant ($p < 0.05$) compared to all the baselines, consistently for all the query budgets.

## A.8    PERFORMANCE OF OUR FRAMEWORK USING THE FAST QP SOLVER

We investigated the literature for potential solutions to improve the computational overhead of the QP solver. We implemented a solution, where the strategy is to solve the QP problem in the GPU (instead of the CPU) using the parallel implementation of the alternating direction method of multipliers, as detailed in Schubiger et al. (2020).

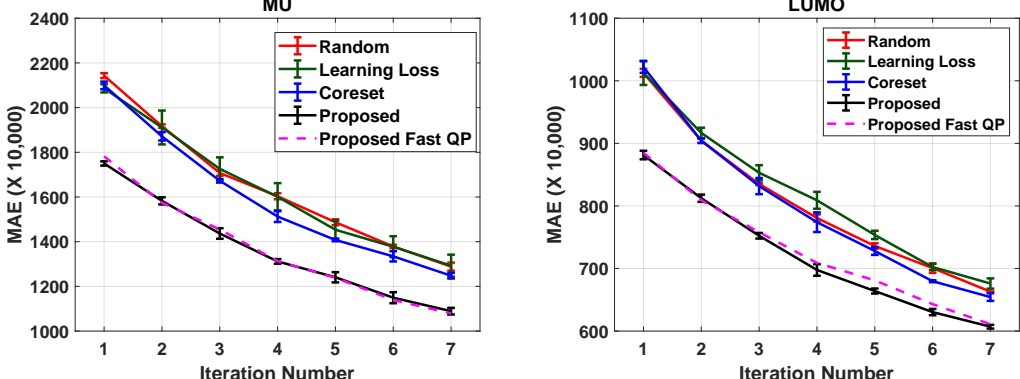

Figure 8: Active learning performance results of the proposed method using the Fast QP solver. Best viewed in color.

We conducted an experiment to validate the performance of this QP solver. The results on the *mu* and *lumo* properties are depicted in Fig. 8. As evident from the figure, the performance of our method using the fast QP solver is very similar to that obtained using the original QP solver. Our method still consistently and comprehensively outperforms all the baselines. However, due to the GPU based QP implementation, solving the QP problem now takes less than 1 minute (with $20,000$ unlabeled molecules). The average time taken for one AL iteration (sample selection and training the SphereNet model) by our method using the fast QP implementation is now 1 hr 28 mins (as opposed to 1hr 58 mins before). This value has been reported in Table 2. Thus, using the fast QP solver, the computation time of our framework has been substantially reduced, with almost similar active learning performance.

