# OpenReview forum: "Empowering Active Learning for 3D Molecular Graphs with Geometric Graph Isomorphism"
_ICLR.cc/2024/Conference — Submitted to ICLR 2024_

### Official Review · Reviewer_wRPn · 2023-10-28

**Soundness:** 2 fair
**Presentation:** 2 fair
**Contribution:** 3 good
**Rating:** 6
**Confidence:** 2

**Summary:**

The authors consider molecular graph classification where the data sets consists of a graph + 3d coordinates modeling a molecule. Thy introduce a set of isometrics which are meant to encode the graphs geometric structure and then use this isometrics to guide active learning on the molecular data set.

**Strengths:**

Able to integrate both spatial and graph information into an active learning pipeline. Promising preliminary numerics

**Weaknesses:**

Proposed method is slower than most of the competing methods because of the need to solve a QP

It would be good to more clearly the entire method (and how all the submodules fit together) a more clear way, e.g., algorithm environment, itemized list, figure.


The proof that your descirtors are stronger than GWL is less impactful because it is ** before ** the computation of moments. The hard part of getting a maximally expressive GNN is an injective aggregation scheme

The authors should make it more clear how \zeta is constructed from the different notions of isometry

The proof of theorem A.1 introduces new notation without explaining it

The proof of theorem 2 (part 2) omits the word ``theorem" while citing theorem 1 and also seems to misquote theorem 1. The proof of theorem 2 says cross angular is the strictest, but thats not what theorem 1 says.

Minor Issue

Sections A.2 should be before A.1. (I want to know what the GWL is BEFORE I read the proofs.)

**Questions:**

Theorem 1 gives relationships between 2 subsets of the isometrics. Can anything be said of the remaining subset?

Could you please provide a more specific reference to the theorem of Hall that you reference (i.e. theorem number and/or page number)

---

> ### Author Response · Authors · 2023-11-16
> **Performed extra experiments with a fast QP Solver; included a schematic diagram for the overall pipeline; compared to GNN's expressiveness; largely improved writing and presentation; clarified some parts (1st part)**
>
> We appreciate your insightful feedback and concerns! **We have revised the manuscript substantially (marked in red) and also provide responses here.**
>
> > **Q: Proposed method is slower than most of the competing methods because of the need to solve a QP.**
>
>
> Thank you for your concern! **In short, even though our main novelty is to design new diversity and uncertainty components for 3D molecular graphs, we implemented a faster QP solver, which reduces the time in Table 2 from 1hr 58 mins (in the original version) to 1hr 28 mins now, with almost the same performance as before. We updated Sec 4.5 and Table 2, and added a new section A.8 in Appendix for detailed analysis.**
>
>
> - First, we would like to point out that, **Our main novelty is to design new diversity and uncertainty components for 3D molecular graphs.**
> Diversity and uncertainty are two typical measures in active learning (AL). They describe which samples should be selected from different perspectives; diversity describes how a subset can contain as much information as the original full set, and uncertainty indicates how the model is confident in each sample. Usually, combining them two would help select the most informative subset for AL. **With uncertainty and diversity components, it's natural to formulate AL to a QP problem. Our Experimental studies (especially the ablation study) also show BOTH components contribute to the performance a lot.**
>
>
> - Studying QP in AL field is not new, and there exist a couple of QP solvers, of which the computational speed varies. As studying QP is not the main purpose and novelty of this work, we employ the commonly used Operator Splitting Quadratic Program (OSQP) [1] to solve the QP problem in Eq. (4). We then use a greedy approach to project the continuous solution back to the binary space, where the $k$ highest entries of the continuous solution vector are set to $1$ and the remaining to $0$. Such an approach is commonly used to convert continuous solutions obtained from a QP solver to binary solutions in AL research [2, 3].
>
>
> - However, we agree that efficiency is a critical consideration in AL research. Hence, we **investigated the literature and implemented a solution, where the idea is to solve the QP problem in the GPU (instead of the CPU)** using the parallel implementation of the alternating direction method of multipliers, as detailed in [4]. We conducted an experiment to validate the performance of this QP solver. The results on the **mu** and **lumo** properties are depicted in `Fig. 8 in Sec. A.8 of the Appendix`. As evident from the figure, the performance of our method using the fast QP solver is very similar to that obtained using the original QP solver. Our method still consistently and comprehensively outperforms all the baselines. However, due to the GPU based QP implementation, solving the QP problem now takes less than $1$ minute (with $20,000$ unlabeled molecules). The average time taken for one AL iteration (sample selection and training the SphereNet model) by our method using the fast QP implementation is now 1 hr 28 mins (as opposed to 1hr 58 mins before). This value has been reported in Table 2 of the paper. Thus, using the fast QP solver, the computation time of our framework has been clearly reduced, with almost the same performance. **As mentioned above, solving QP problem is not our main novelty. So we believe the novelty of our work should be evaluated based on the current state.** We will continue exploring even faster QP solvers from the literature for engineering purposes as future work.
>
>
>
> [1] B. Stellato, G. Banjac, P. Goulart, A. Bemporad, and S. Boyd. OSQP: an operator splitting solver for quadratic programs. Mathematical Programming Computation, 12(4):637-672, 2020. doi: 10.1007/
> s12532-020-00179-2. URL https://doi.org/10.1007/s12532-020-00179-2
>
>
> [2] Rita Chattopadhyay, Wei Fan, Ian Davidson, Sethuraman Panchanathan, and Jieping Ye. Joint transfer and batch-mode active learning. In International Conference on Machine Learning (ICML), 2013.
>
>
> [3] Zheng Wang and Jieping Ye. Querying discriminative and representative samples for batch mode active learning. In ACM SIGKDD International Conference on Knowledge Discovery and Data Mining, 2013.
>
>
> [4] Michel Schubiger, Goran Banjac, and John Lygeros. GPU acceleration of ADMM for large-scale quadratic programming. Journal of Parallel and Distributed Computing, 144: 55-67, 2020

---

> ### Author Response · Authors · 2023-11-16
> **Performed extra experiments with a fast QP Solver; included a schematic diagram for the overall pipeline; compared to GNN's expressiveness; largely improved writing and presentation; clarified some parts (2nd part)**
>
> > **Q: It would be good to more clearly the entire method (and how all the submodules fit together) a more clear way, e.g., algorithm environment, itemized list, figure.**
>
> Thank you for the valuable suggestion.
> We have included a schematic diagram `(Fig. 6) and described it in Sec. A.4` of our Appendix, which shows how all the submodules of our framework fit together.
>
>
> > **Q: The proof that your descriptors are stronger than GWL is less impactful because it is ** before ** the computation of moments. The hard part of getting a maximally expressive GNN is an injective aggregation scheme**
>
> Thank you for your comment.
>
> - Regarding the first point, we explain in two parts: what's happening **before** the computation of moments, and what's happening **after** involving moments.
>   - **Before:** When constructing $GR_{\text{ours}}$, our focus is - to find which geometries can fully describe a complete isometry space. Based on our theory study on isometrics, we conclude $GR_{\text{ours}}$ should include all reference distances, triangles, and cross angles in a 3D graph (We have added several statements under Theorem in Sec 2.1.2 for clear presentation). We then prove $GR_{\text{ours}}$ has the greater expressive power than the GWL Test. **This conclusion is still significant, as there could be many ways to transform $GR_{\text{ours}}$ into some representations that can serve as direct inputs to models. People can research their ways based on our $GR_{\text{ours}}$ for complete molecular structural representations.** We thus believe this point is a significant contribution to the community.
>   - **After:** In this work, we tackle this from a distribution perspective, as the previous well-known work USR [1] (for molecular shape analysis) leverages this distribution perspective and achieves very successful outcomes. For sure, we agree - there could be a doubt that using moments may not fully reconstruct the distribution (either in the USR paper or our work). However, **as will be analyzed when answering your last question**, we can reconstruct the distribution with high fidelity by increasing the number of translated moments as well as the order of computed moments. We believe this can serve for precise characterization of molecular structures and thus allows for a good diversity calculation. Our impressive experimental results can also support the effectiveness of this solution.
>
>
> - For the second point, indeed, as you said, the design of the injective aggregation scheme of a GNN is the core to give a better expressive power of the GNN. And it’s also true that we *could* use a GNN to learn a molecular structure representation for diversity calculation. However, the main purpose of this research is NOT to design a powerful GNN model, but to design a way to compute diversities between two 3D molecules of any size for active learning. For sure, a perfectly pretrained GNN model may distinguish two molecules at a reasonably good accuracy level, but we cannot guarantee this in practice. Furthermore, there could be domain shift issues (pretrained on one data and perform on another data with different distribution) that harm the active learning performance. Hence, we design a `model-agnostic` solution by directly performing isomorphy study on 3D molecular conformations, `avoiding the need of perfectly pretrained GNN models, as well as achieving a guaranteed upper bound of the expressiveness of all existing models`. **In the original submission, we had a paragraph at the end of Sec 2.1.2 to explain this. Based on your comments, we added more statements for further clarification.**
>
>
>
> > **Q: The authors should make it more clear how \zeta is constructed from the different notions of isometry.**
>
> Thank you for the suggestion.
>
> - First, for rigorous presentation, in our paper, $GR$ denotes a geometric representation (like a set of distances and cross angles) to capture molecular structures; a function $\zeta$ denotes the embedding function associated with a $GR$, such that $\zeta$ takes a point cloud as input to produce a $GR$.
>
>
> - Through analysis and theoretical studies in Sec 2.1.1,
> we naturally conclude that, the proposed $GR_{\text{ours}}$ should include reference distances, triangles, and cross angles.  By doing this, $GR_{\text{ours}}$ has greater expressive power than the GWL Test, so it can precisely capture the 3D shape diversity among different molecules. **We have added several statements under Theorem 2 in Sec. 2.1.2 of the paper to clarify how to construct $GR_{\text{ours}}$.**
>
>
>
> [1] Pedro J Ballester and W Graham Richards. Ultrafast shape recognition to search compound databases for similar molecular shapes. Journal of computational chemistry, 28(10):1711–1723, 2007.

---

> > ### Comment · Reviewer_wRPn · 2023-11-22
> >
> > In light of the reviewers thorough response, I have increased my rating.

---

> ### Author Response · Authors · 2023-11-16
> **Performed extra experiments with a fast QP Solver; included a schematic diagram for the overall pipeline; compared to GNN's expressiveness; largely improved writing and presentation; clarified some parts (3rd part)**
>
> > **Q: The proof of theorem A.1 introduces new notation without explaining it.**
>
> Thank you for pointing this out, and we have revised the proof to address these concerns. Specifically, we have included explicit explanations for the symbols $\Delta$ and $d(r_{2}, f(a))$, which previously lacked clear definitions. In the new proof, we have made sure to use defined symbols within the context of the theorem, ensuring their meanings are unambiguous and directly tied to the concepts being discussed. We believe these changes will make our presentation much clearer as you suggested.
>
>
>
>
>
>
>
>
> > **Q: The proof of theorem 2 (part 2) omits the word ``theorem" while citing theorem 1 and also seems to misquote theorem 1. The proof of theorem 2 says cross angular is the strictest, but that's not what theorem 1 says.**
>
>
> Thank you for pointing this out. You are right that we had misquoted Theorem 1 in the proof of Theorem 2. The constraints on $GR$ in Theorem 2 are already enough for the proof.
> **We have updated the proof located in Sec. A.2 in Appendix.**
>
>
> > **Q: Sections A.2 should be before A.1. (I want to know what the GWL is BEFORE I read the proofs.)**
>
>
> Thank you for your suggestion!
> We have adjusted the order of the sections so that the introduction to the Geometric Weisfeiler-Leman (GWL) Test is now presented prior to the proofs.
> We believe this change enhances the readability and logical flow of the paper, and we appreciate your guidance in making this improvement!
>
>
> > **Q: Theorem 1 gives relationships between 2 subsets of the isometrics. Can anything be said of the remaining subset?**
>
>
> - Originally, we didn't include a direct relationship between Cross Angular Isometric and Triangular Isometric in Theorem 1, but we created `Fig. 2` to visually articulate this relationship.
>
>
> - Taking your insightful advice into account, we've revisited the theorem to clarify the connection. Now it should paint a clearer picture of how all the isometrics relate to each other. Really appreciate you pointing this out — it's definitely tightened up our work.

---

> ### Author Response · Authors · 2023-11-16
> **Performed extra experiments with a fast QP Solver; included a schematic diagram for the overall pipeline; compared to GNN's expressiveness; largely improved writing and presentation; clarified some parts (4th part)**
>
> > **Q: Could you please provide a more specific reference to the theorem of Hall that you reference (i.e. theorem number and/or page number)**
>
>
> Thank you for the question.
>
>
> - Specifically, Theorem 1 in Hall’s paper [1] was what we referred to. In Theorem 1, the author states that *"for a random variable $X$ with a finite $p^\text{th}$ moment where $p$ is not an even integer, the distribution of $X$ is uniquely characterized by the sequence of its translated moments"*. Inspired by this theorem, we define `four reference points to` obtain four **"translated"** geometries for each of the three geometry in $GR_{\text{ours}}$ (reference distance, triangle, cross angle).
> Take the cross angle as an example, we use four reference points to compute all cross angles, including the centroid, the point closest to the centroid, the point farthest from the centroid, and the point farthest from fct. **These four reference points are expected to capture the "translated" cross angles**. For each reference point, we compute a set of moments. As stated in Theorem 1 in [1], $p$ is not an even integer. Following the USR work [2] that suggests $p$ =3 (3rd moment) should be enough to uniquely characterize the distribution. Also, the USR work [2] uses other moments to enhance the representation (1st and 2nd moments). Following the thread and based on the experimental results (the setting with the best performance), we finally use the first four moments (mean, variance, skewness, and kurtosis) for each reference point as a comprehensive moment representation (meets Theorem 1 in [1] for sure).
> As a result, we obtain a sequence of translated moments to represent cross angle distribution (a `16-dim vector`: 4 moments for each reference point and we have 4 reference points). We repeat the process for reference distance and triangle distribution, and obtain a `48-dim vector finally`. **Since the sequence of translated moments can be used to reconstruct the distributions of all the three geometries in $GR_{\text{ours}}$, and $GR_{\text{ours}}$ is more powerful than GWL Test, we believe our final representation can capture essential characteristics of a molecule’s shape.** Our strong empirical studies also support this.
>
>
> - As mentioned above, we also referred to the USR paper [2] when working on the manuscript. This work uses distance information only and employs the first three moments (mean, variance, skewness) to approximate the distribution of distances. The authors have a statement in the paper (5th pape, right column, 2nd paragraph) *'Such an approach is based on a theorem [1] from statistics, which proves that a distribution is completely determined by its moments. In this work, we consider the first three moments of each distribution because they provide an excellent compromise between the efficiency and effectiveness of the method'*. This is an impactful work on molecular shape similarity. Both the authors (Pedro Ballester at ICL and Graham Richards at Oxford) are well-known experts in computer-aided molecular design. Based on their statements, we found distance information alone cannot delineate a complete isometry space. This motivated us to study graph isomorphy for a complete isometry space, eventually being a theoretically guaranteed solution for precise molecular diversity computing.
>
> - **We have substantially revised the first paragraph of 2.1.3 to make the presentation clearer.**
>
>
> [1] Hall P. A distribution is completely determined by its translated moments. Zeitschrift für Wahrscheinlichkeitstheorie und Verwandte Gebiete. 1983.
>
>
> [2] Pedro J Ballester and W Graham Richards. Ultrafast shape recognition to search compound databases for similar molecular shapes. Journal of computational chemistry, 28(10):1711–1723, 2007.

---

> ### Author Response · Authors · 2023-11-22
>
> Dear reviewer wRPn,
>
> Thanks for acknowledging our work and our efforts in addressing the comments from reviewers!

---

### Official Review · Reviewer_6CA5 · 2023-10-30

**Soundness:** 3 good
**Presentation:** 3 good
**Contribution:** 3 good
**Rating:** 6
**Confidence:** 4

**Summary:**

The paper introduces an active learning pipeline for molecular learning by viewing molecules as 3D geometric graphs. A notable contribution is the introduction of 3D graph isometrics for 3D graph isomorphism. The authors provide theoretical proof demonstrating the superiority of the proposed method over the existing geometric WL test. A Bayesian geometric GNN is proposed and integrated into the active learning framework to compute uncertainties, which is accomplished by leveraging the estimated moments of the distributions of the 3D geometrics. The active sampling is formulated as a quadratic programming problem. Experiments on QM9 dataset are conducted to demonstrate the effectiveness of the proposed method.

**Strengths:**

+ The introduction of 3D graph isometrics in this paper is both intriguing and novel. Despite its deep theoretical roots in group theory, the application to the problem of active learning is innovative and well justified. The authors have done a good job in detailing the theoretical proofs of the proposed graph isometrics, alongside a comprehensive comparison with existing methodologies like the geometric WL test.

+ The paper is well-written and articulate. While there’s room for enhancement (see questions and weakness), the current version of the manuscript efficiently facilitates a clear understanding of both the overarching motivation and the technical details in the proposed methodology.

+ The experimental results presented are compelling, effectively underscoring the practical efficacy and reliability of the proposed method.

**Weaknesses:**

One aspect that appears to be lacking in the method section is a detailed discussion on the motivation behind choosing the specific three 3D isometrics proposed. While it’s evident that these constitute a complete isometry space, there exist various alternative approaches and descriptors to delineate the isometry space, such as using the first and second fundamental forms. Enhancing the section with additional references and a more thorough discussion that delves into the rationale behind the selection of these particular 3D isometrics would significantly bolster the clarity and comprehensiveness of the presentation.

**Questions:**

- The paper keeps using "2D graph/molecule" to refer to "planar graph/molecule" or "topological graph". It is not quite concise, as "2D" already indicates geometric information is considered

- \mathcal{SE}(3) -> $SE(3)$

- Page 2 Section 2.2.1, how to ensure the bijective function f always exists? What if A and B have different numbers of points?

- Page 2 Section 2.2.1, how do you define A and B? I guess you are using the matrix by concatenating the point positions, but an explicit definition is required.

- Page 5 Section 2.3, what is the specific "off-the-shelf QP solver" being used? It is important to reveal the info here since different frameworks can result in very different computational speeds, as is already demonstrated in the experimental section. For example, how do you project the continuous solution to the binary space, are you using branch-and-bound?

- For the experiments, I noticed almost all the cases, the proposed method achieves a better performance than baselines in a large margin at the beginning of the iteration. Though it can be attributed to the proposed GNN, I was wondering what will happen if the proposed method starts from a "bad" set of initial samples. How robust is the proposed method to the initialization? Can it still converge to a better result if the initialization is worse than existing methods?

---

> ### Author Response · Authors · 2023-11-16
> **Added detailed discussion on the motivation for a complete isometry space; largely improved writing and presentation; clarified some parts (1st part)**
>
> We thank you for saying our work is novel, well-written, and empirically strong. We particularly appreciate your insightful feedback and concerns! **We have revised the manuscript substantially and also provide responses here.**
>
> > **Q: One aspect that appears to be lacking in the method section is a detailed discussion on the motivation behind choosing the specific three 3D isometrics proposed. While it's evident that these constitute a complete isometry space, there exist various alternative approaches and descriptors to delineate the isometry space, such as using the first and second fundamental forms. Enhancing the section with additional references and a more thorough discussion that delves into the rationale behind the selection of these particular 3D isometrics would significantly bolster the clarity and comprehensiveness of the presentation.**
>
> Thank you for your insightful feedback! **Based on the suggestion, we added a paragraph at the end of Sec 2.1.1 to provide a detailed rationale for developing the three 3D isometrics.**
> In short,  the three 3D isometrics - Reference Distance, Triangular, and Cross Angular - were developed for `their natural use in molecular shape analysis and high computational efficiency`.
>
> - First, our objective is to develop an accurate method for molecular diversity computing.
> Existing mainstream paradigms for molecular diversity, like USR and Gaussian overlay-based methods (references see Related Work Sec 3.2 in paper), are all based on geometries (distances, angles) in Euclidean space. Through analysis in Sec 2.1.1 and 2.1.2, we can easily see our methods will render several geometries, including reference distances, triangles, and cross angles. Thus, our three 3D isometrics can be **naturally and easily integrated into existing pipelines for accurate molecular diversity computing**.
>
> - Second, our method is more computationally efficient.
>   - The mentioned *first and second fundamental forms* indeed provide a detailed capture of a surface's intrinsic and extrinsic geometries. By definition, the first fundamental form is the inner product on the tangent space of a surface, and the second fundamental form is a quadratic form on the tangent plane of a surface. However, they require the computation of differential properties such as gradients and curvatures from a point cloud, which is a discrete representation of a molecule's structure. When applied to non-smooth surfaces, such as molecular point clouds, this can lead to a substantial increase in computational effort. For instance, to approximate these forms, one might have to fit a local surface at every point and compute its curvature using second-order derivatives. If this process involves examining the interactions of every triplet of points to fit the surface, the complexity can escalate to $O(N^3)$ ($N$ is the number of points), which becomes computationally intensive for large molecules.
>
>   - In contrast, the three isometrics we employ are designed for direct computation from point cloud data. The **Reference Distance Isometric** and **Triangular Isometric** both have a computational complexity of $O(N)$, as they involve operations over $N-1$ points. The **Cross Angular Isometric** has a complexity of $O(N^2)$, considering the angles between all pairs of points, which, while higher, is still significantly more tractable than $O(N^3)$.
>
>   - In short, our approach avoids the computational burden, such that we maintain a tradeoff between descriptive power and computational feasibility for complete isometry space. This is crucial for active learning tasks, where high-throughput and computational speed are also paramount concerns.
>
>
> > **Q: The paper keeps using "2D graph/molecule" to refer to "planar graph/molecule" or "topological graph". It is not quite concise, as "2D" already indicates geometric information is considered.**
>
> Thanks for the value point. We agree that the word "2-dimensional" somehow infers geometric information. We think "planar graph/molecule" is truly precise, thus we changed "2D graph" to "planar graph" and  "2D molecule" to "planar molecule" in the revised version.
>
> > **Q: \mathcal{SE}(3) -> SE(3)**
>
> Thanks for pointing this out. We have changed accordingly in the paper.

---

> ### Author Response · Authors · 2023-11-16
> **Added detailed discussion on the motivation for a complete isometry space; largely improved writing and presentation; clarified some parts (2nd part)**
>
> > **Q: Page 2 Section 2.2.1, how to ensure the bijective function f always exists? What if A and B have different numbers of points?**
>
> Yes, you are right, we can not ensure that the bijective function $f$ always exists. The mapping $f$ is a key element to check if a method can pass three levels of isometrics, to evaluate the expressiveness of the method in distinguishing different geometric graphs (this is our main research purpose to develop a good diverse method). Clearly, when $A$ and $B$ have different numbers of points, it's very easy to distinguish them. That is also to say, if there does not exist such an $f$ for a method M, this method M can not even distinguish two point clouds with different points. Then M has very poor expressiveness and is not even needed to pass the three isometrics. The method M then can NOT be a good diversity method.
>
> In this work, we introduce three levels of isometrics to examine the expressive power of different methods. Our proposed $GR_{\text{ours}}$ can pass all three levels of isometrics, so it can distinguish two similar molecules. As a result, $GR_{\text{ours}}$ is a good diversity method (with a theoretical guarantee).
>
>
> > **Q: Page 5 Section 2.3, what is the specific "off-the-shelf QP solver" being used? It is important to reveal the info here since different frameworks can result in very different computational speeds, as is already demonstrated in the experimental section. For example, how do you project the continuous solution to the binary space, are you using branch-and-bound?**
>
> We agree that the computational speed of different QP Solvers varies.
> In this work, the Operator Splitting Quadratic Program (OSQP) [1] is used to solve the QP problem in Eq (4). We then employ a greedy approach to project the continuous solution back to the binary space, where the $k$ highest entries of the continuous solution vector are set to $1$ and the remaining to $0$. Such an approach is commonly used to convert continuous solutions obtained from a QP solver to binary solutions in AL research [2, 3].
>
> **We have updated section 2.3 to include this text and the references to address your comment.**
>
> [1] B. Stellato, G. Banjac, P. Goulart, A. Bemporad, and S. Boyd. OSQP: an operator splitting solver for quadratic programs. Mathematical Programming Computation, 12(4):637-672, 2020. doi: 10.1007/
> s12532-020-00179-2. URL https://doi.org/10.1007/s12532-020-00179-2
>
> [2] Rita Chattopadhyay, Wei Fan, Ian Davidson, Sethuraman Panchanathan, and Jieping Ye. Joint transfer and batch-mode active learning. In International Conference on Machine Learning (ICML), 2013.
>
> [3] Zheng Wang and Jieping Ye. Querying discriminative and representative samples for batch mode active learning. In ACM SIGKDD International Conference on Knowledge Discovery and Data Mining, 2013.

---

> ### Author Response · Authors · 2023-11-16
> **Added detailed discussion on the motivation for a complete isometry space; largely improved writing and presentation; clarified some parts (3rd part)**
>
> > **Q: For the experiments, I noticed almost all the cases, the proposed method achieves a better performance than baselines in a large margin at the beginning of the iteration. Though it can be attributed to the proposed GNN, I was wondering what will happen if the proposed method starts from a "bad" set of initial samples. How robust is the proposed method to the initialization? Can it still converge to a better result if the initialization is worse than existing methods?**
>
> Thank you for the comments. We think there are some misunderstandings.
>
> - The performance of the proposed method (or any active learning algorithm, for that matter) is dependent on the initial set of labeled samples. A “bad” set of initial samples will affect the performance of our AL algorithm, particularly, in the first few AL iterations; however, once we have a good set of samples in the labeled set (after some AL iterations), our method will start querying the informative samples and given a sufficient number of AL iterations, it will still be able to converge to a good result.
>
> - This argument is true for any AL algorithm; thus, from the trend in the results in Figure 3 in our paper, we can conclude that a “bad” set of initial samples will affect the baseline methods even more severely, and our method will still maintain a large margin of performance difference compared to the baselines.
>
> - Additionally, for fair comparisons, the initial labeled data samples and the backbone 3D graph model should remain consistent across all active learning methods. As the backbone model is the same, a better active learning framework will select better samples for annotation, and then it should still perform better on the initial data.
> If our method performs **badly** on the initial data, then the other models must perform **worse**. Then, **a good active learning method should be better than others from the initial stage and iteration 1**. The number of iterations is dependent on the annotation budget. "Convergence" is not so important in active learning, where the **main purpose is to develop an AL model that is the best for any budget**.
>
> Hope these comments can address your concerns. We would like to discuss more if you have other questions.

---

> > ### Comment · Reviewer_6CA5 · 2023-11-20
> >
> > Thanks for the authors' rebuttal. It addresses all my concerns and I will keep my positive rating.

---

### Official Review · Reviewer_TZEh · 2023-10-31

**Soundness:** 2 fair
**Presentation:** 2 fair
**Contribution:** 3 good
**Rating:** 6
**Confidence:** 2

**Summary:**

This paper studies active learning (AL) for 3D geometric modeling for molecules. It introduces a diversity matrix to augment the active sampling in the AL process and reaches good empirical performance.

**Strengths:**

- The problem of active learning is an important and novel task in AI for drug discovery.
- The experiments seem solid.

**Weaknesses:**

- Some claims need to be corrected.
    - In the first paragraph in Sec 1, the 3D graph usually does not include chemical bonds.
    - In the third paragraph in Sec 1, it should be SE(3)-equivariant, not SE(3)-invariant. If the model is SE(3)-invariant (as it is now), then the claim that “can be applied in conjunction with any 3D GNN architecture” is not correct because there are equivariant models not included in invariant modeling.

- The core method needs more clarity to be understood:
    - In Sec 2.1.1, what does “equivalent” mean in “$h_{f(a)}$ is equivalent to $h_a$”? Does it mean identical? Equivariant (if so, there’s no such a statement for equivariance)? Or sth else?
    - In Sec 2.1.1, what do the curly brackets mean in “We define A to be mapped {-Reference Distance, … } isometrically to B, if …”? I can understand this sentence if without the curly brackets. More clarifications are needed here.
    - In Sec 2.1.1, what does “together with an additional constraint for each scenario” mean?
        - This sentence is saying, “A is isometric to B, if … s.t. …”. Then the authors say that “and with additional constraint for two scenarios: (1) If …, s.t. …; (2) If …, s.t. …; (3) If …, s.t.”
        - With all due respect, I do not fully understand what these sentences mean.
        - I can guess what the authors mean from the context, but such a description is not good enough and needs polishment.

- Almost all the equivariant geometric models (SchNet, EGNN, SphereNet, GemNet, TFN, Equiformer, etc) satisfy the ‘isometric’ condition. I am not sure why the authors highlight this in Sec 2.1.1 - 2.1.2. Besides, Sec 2.1.1-2.1.2 are disconnected from Sec 2.1.3 and Sec 2.2.

**Questions:**

See above.

---

> ### Author Response · Authors · 2023-11-16
> **Largely improved writing and presentation; clarified some parts; explained the relationship to equivariant geometric models (1st part)**
>
> Many thanks for your valuable comments. **We have revised the manuscript heavily (marked in red) to improve the writing and presentation, and we also provide responses here.**
>
> > **Q: In the first paragraph in Sec 1, the 3D graph usually does not include chemical bonds.**
>
> Thank you for pointing this out. Yes, you are right. Most existing 3D datasets do not contain information of chemical bonds. Based on the research purpose, different research works use different ways to determine the connections among atoms (if two atoms are connected or not). In most existing 3D GNN studies (SchNet, DimeNet, SphereNet, etc), when constructing 3D graphs, the connectivities among atoms are usually based on a predefined `cut-off distance`. For example, if we set the cut-off distance to be 6A as a threshold, then if the Euclidean distance between two atoms is smaller than or equal to 6A, there will an an edge between them in the 3D graph. Usually, edge information is critical as the message will be passing through edges (there is no message passing between two nodes if there is no edge between them).
>
> **We have updated the first paragraph in Sec 1 to clarify this.**
>
>
> > **Q: In the third paragraph in Sec 1, it should be SE(3)-equivariant, not SE(3)-invariant. If the model is SE(3)-invariant (as it is now), then the claim that “can be applied in conjunction with any 3D GNN architecture” is not correct because there are equivariant models not included in invariant modeling.**
>
> Thank you for the valuable suggestion. The output of our proposed diversity computing module is a set of geometries, like distances and angles. Hence, our module can be incorporated into any existing equivariant geometric model (SchNet, EGNN, SphereNet, GemNet, TFN, Equiformer, etc).
>
> **We have corrected this in the third paragraph in Sec 1.**
>
> > **Q: In Sec 2.1.1, what does "equivalent" mean in " $h_{f(a)}$ is equivalent to $h_a$ "? Does it mean identical? Equivariant (if so, there's no such a statement for equivariance)? Or sth else?**
>
> Thank you for bringing this to our attention. The intended meaning was indeed `identical`, and what was in the paper was a typo.
> We have updated to $h_{f(a)} = h_a$ in the first paragraph of Sec. 2.1.1.
>
> > **Q: Confusion on texts from “We define A to be mapped {-Reference Distance, … } isometrically to B” to  “together with an additional constraint for each scenario”.**
>
> We admit that these statements in the original version are confusing, so we have revised this part in the paper.
>
> In short, the statement *“There needs to exist a bijective function $f: A \rightarrow B$, such that $h_{f(a)} = h_a$ for any node $a \in A$.”* is a **common and minimal requirement** for all three isometrics. Each isometric then has **additional constraints**.
>
> Specifically, we define three isometrics, including Reference Distance Isometric, Triangular Isometric, and Cross Angular Isometric. Each of the three isometrics is associated with several constraints. However, the above statement regarding a mapping $f$ is common for all three isometrics. You can also easily find that, in the definition of each isometric, we need such a mapping $f$. In summary, the above statement is the **common and minimal** requirement for all three isometrics. Hope the revised version is much clearer, and please let us know if you find any further issues regarding this.

---

> ### Author Response · Authors · 2023-11-16
> **Largely improved writing and presentation; clarified some parts; explained the relationship to equivariant geometric models (2nd part)**
>
> > **Q: Almost all the equivariant geometric models (SchNet, EGNN, SphereNet, GemNet, TFN, Equiformer, etc) satisfy the 'isometric' condition. I am not sure why the authors highlight this in Sec 2.1.1 2.1.2. Besides, Sec 2.1.1-2.1.2 are disconnected from Sec 2.1.3 and Sec 2.2.**
>
> Thanks for starting this insightful discussion!
>
>
> - First, for the relationship between our method and existing equivariant geometric models - In short, our method is a `model-agnostic` solution by directly performing isomorphy study on 3D molecular conformations, **avoiding the need of a perfectly pretrained equivariant geometric model, as well as achieving a guaranteed upper bound of the expressiveness of all existing geometric models.**
>
>   - The contribution of 'isometric' condition is that, we set an upper bound (with theoretical guarantee) of the expressiveness of ALL existing equivariant geometric models. Actually, we define isometrics at three levels of expressiveness. For example, a **well pretrained SchNet** (only considers distance information) is upper bounded by Reference Distance Isometric (but not Triangular Isometric or Cross Angular Isometric); a **well pretrained DimeNet** (more powerful than SchNet) is upper bounded by Triangular Isometric (but not Cross Angular Isometric).
>
>   - The main purpose of this research is NOT to design a powerful equivariant geometric model, but to design a way to compute diversities between two 3D molecules of any size for active learning. For sure, a perfectly pretained equivariant geometric model may distinguish two molecules at a reasonably good accuracy level, but we can not guarantee this in practice. Furthermore, there could be domain shift issues (pretrained on one data and perform on another data with a different distribution) that harm the active learning performance. Hence, we design a `model-agnostic` method by directly performing an isomorphy study on 3D molecular conformations, avoiding these risks, as well as achieving a guaranteed upper bound of all existing equivariant geometric models.
>
>   - **In the original submission, we had a paragraph at the end of Sec 2.1.2 to explain this. Based on your comments, we added more statements for further clarification.**
>
> - Second, the relationships among several sections.
>
>   - Regarding the relationships among Sec 2.1.1, Sec 2.1.2, and Sec 2.1.3:
>   In short, through analysis and theoretical studies in **Sec 2.1.1**, we conclude $GR_{\text{ours}}$ should include all reference distances, triangles, and cross angles in a 3D graph in **Sec 2.1.2**. We also formally prove $GR_{\text{ours}}$ has greater expressive power than the GWL Test in **Sec 2.1.2**, so it can precisely capture the 3D shape diversity among different molecules. Then the final distributional representations in **Sec 2.1.3** are based on these three geometries expressed in $GR_{\text{ours}}$. **We have added several statements under Theorem 1 in Sec 2.1.2, as well as in the first paragraph of Sec 2.1.3 to clearly indicate such relationship.**
>
>   - Regarding the relationships between Sec 2.1 and Sec 2.2: Diversity and uncertainty are two typical and different measures in active learning. **Sec 2.1 focuses on diversity and Sec 2.2 focuses on uncertainty.** These two components describe which samples should be selected from different perspectives; diversity describes how a subset can contain as much information as the original full set, and uncertainty indicates how the model is confident in a sample. Usually, combining them two would help select the most informative subset for active learning. **Our main novelty is designing new diversity and uncertainty components for 3D graphs.** The Experimental studies (especially the ablation study) also show both components contribute to the active learning performance a lot. **We have added several statements in the first paragraph of Sec 2.2 to clarify this.**

---

> ### Author Response · Authors · 2023-11-21
> **Dear reviewer TZEh: could you check our response?**
>
> Dear reviewer ` TZEh`: Thank you for your suggestions regarding the paper's writing.  We have corrected all statements as you advised, and clarified all the parts you mentioned. Please check our response and paper revision and let us know if there are further comments. Otherwise, we would greatly appreciate it if you could increase your score. Thank you.

---

> > ### Comment · Reviewer_wRPn · 2023-11-21
> > **Acknowledgement of response**
> >
> > I am writing to acknowledge the authors responses and thank them for clarifying things. However, I unfortunately still feel that this paper is slightly below the acceptance threshold.

---

> > ### Comment · Reviewer_TZEh · 2023-11-22
> > **A Quick Confirmation**
> >
> > Hi authors,
> >
> > I am still reading your replies.
> >
> > Can you help quickly explain what you mean by `The main purpose of this research is NOT to design a powerful equivariant geometric model, but to design a way to compute diversities between two 3D molecules of any size for active learning`?
> >
> > I am asking this for two reasons:
> > - The word `representation` has been widely used in the literature, which is equivalent to geometric modeling.
> > - Another question is how you design the diversity? Is this still based on the representation space?

---

> ### Author Response · Authors · 2023-11-22
> **Reply to reviewer TZEh**
>
> Dear reviewer TZEh,
>
> Thanks for the question. The reason why we said the main purpose is not to design a powerful equivariant model is - We can use ANY equivariant geometric model as a backbone network, and in this work we use SphereNet in experiments.
>
> In short, a representation space is not necessarily to be a learning representation space. Diversity is still in the isometry representation space (without learning), but directly computed from 3D graph isomorphism (also considering domain knowledge which is the molecular shape). `The purpose of the diversity component is still for geometric modeling.`
>
> We have two components: diversity and uncertainty. Uncertainty is to describe - how the geometric learning model is confident about the sample. So we develop a Bayesian learning model (BGGNN) based on SphereNet for learning uncertainty.
>
> Diversity is still based on the representation space. It’s **not** based on the learning representation space (through a geometric learning model).  **It’s in the isometry representation space, and to develop isometric representations without learning.** The output of the diversity component is still - a set of geometric representations. The benefit is - no matter whether the used geometric learning model is powerful or not, and no matter whether the used geometric learning model is well trained or not, **our diversity based on the isometry representation space can be better than any other methods ( learning or non-learning) for geometric modeling** with theoretical guarantees. As a comparison, `the baseline Coreset actually employs the used equivariant geometric learning model SphereNet to learn a vector representation for each molecule to compute the diversity.` We can easily see from the experiments that, our proposed diversity method is much better than Coreset, which again demonstrates the effectiveness of geometric modeling in the isometry representation space.  (It's similar to that - people compute ground truth for learning models directly from data's properties, and ground truth is for sure the upper bound of any learning model. Our computing is performed in 3D isometry representation space, and the efficiency is very high (see Table 2 in experiments); Additionally, different levels of isometrics set the upper bound to different equivariant learning models based on their expressiveness power. )
>
> The diversity and uncertainty are combined together to select informative samples (the most critical step for active learning). For performance study, we definitely need equivariant learning models for property prediction.
>
> Our newly added Figure 6 in the appendix as well as the revised texts in the last paragraph of Sec 2.1.2 can also help clarify your question.
>
>
> Thanks,
>
> Authors

---

> ### Author Response · Authors · 2023-11-23
> **Follow up with reviewer TZEh**
>
> Dear reviewer TZEh,
>
> The Discussion period will end in a few hours. Could you let us know if your initial comments have been all addressed and your new question is well answered? Thank you!
>
> Sincerely,
>
> Authors

---

> ### Comment · Reviewer_TZEh · 2023-11-23
> **Reply**
>
> Hi authors,
>
> I am trying my best to understand the paper.
>
> Unfortunately, the first version of the draft lacks too many details/clarifications (and ICLR only allows 2-week rebuttal this year). They are solved now, which I appreciate, and I need to re-read your paper to understand better.

---

> > ### Author Response · Authors · 2023-11-23
> > **Thank you reviewer TZEh!**
> >
> > Dear reviewer TZEh,
> >
> > Absolutely and totally understandable. We really appreciate you being so responsible. Please raise any quick confirmation if have and we will try our best to respond immediately.
> >
> > Thanks,
> >
> > Authors

---

### Author Response · Authors · 2023-11-20
**To all reviewers: Could you please reply to our rebuttal considering the Rebuttal/Discussion phase is ending?**

Dear Reviewers,

Thank you for your constructive comments and suggestions. Our point-by-point responses can be found below. We also have **added more experiments** and **revised our paper heavily** based on your suggestions and comments. We think the presentation is largely improved. Since it is approaching the end of the Rebuttal/Discussion phase and the current rating is borderline, we hope that you can reply to our rebuttal and consider updating your scores if we have addressed your concerns. Also, please let us know if there are any additional concerns or feedback. Thank you!

Sincerely,

Authors

---

### Comment · Area_Chair_SPxi · 2023-11-20
**To all reviewers: Please respond to the authors' rebuttal**

Dear reviewers,

The window for interacting with authors on their rebuttal is closing on Wednesday (Nov 21st). Please respond to the authors' rebuttal as soon as possible, so that you can discuss any agreements or disagreements. Please acknowledge that you have read the authors' comments, and explain why their rebuttal does or does not change your opinion and score.

Many thanks,

Your AC

---

> ### Author Response · Authors · 2023-11-20
>
> Thanks, dear area chair!
>
> Dear review `6CA5`: Thanks for evaluating our work to be intriguing, novel, well-written, and empirically strong. Your major comments were the motivation and some suggestions in writing. We have revised the paper and incorporated all your comments.
>
> Dear reviewers `TZEh` and `wRPn`: We feel your major concerns were the presentation. We have revised the paper substantially to address all your concerns.
>
> Additionally,  reviewer `wRPn` has questions about the QP solver and the theory part (isometry). We provided detailed responses, added experiments, and revised the paper accordingly.
>
> Overall, we really appreciate all the comments, which indeed helped improve the presentation significantly. Technically, our method is among the very early try to design active learning solutions for 3D molecules, which is important for molecular ML and AI4Science (a lot of scientific data can be formulated as 3D graphs, and data is hard to obtain thus the data scarcity is very common in sciences). Our two components are novel. We hope you can take a chance to check if we have addressed all your concerns, and let us know if you have further comments.
>
> Thanks,
>
> Authors

---

### Meta-Review · Area_Chair_SPxi · 2023-12-07

**Metareview:**

The AC would like to thank the authors and reviewers for engaging in constructive discussions. Although the authors and reviewers have engaged in discussions in the reviewing phase, and two scores have been updated, the scores remain fairly borderline. Two out of three reviewers indicated the need for improved clarity on several important parts of the paper. The proposed active learning method was evaluated on QM9 and shows improvements over the baseline methods in terms of MAE. The speed of the method is bottlenecked by the QP part of method, leading to the method being slower than some baselines but faster than one other. The AC commends the authors for doing elaborate work to address the reviewers' concerns, which include extensive revisions on clarity, changes to proofs of theorems, as well as the new experiments that have been done to improve the speed of the method. However, the AC is of the opinion that it is to the benefit of the scientific community that these important revisions undergo a completely new review round, and that the rebuttal period in this conference is too short for reviewers to confidently judge these changes. Therefore, the AC recommends to reject the paper for this conference, and encourages the authors to submit for review at a future venue.

**Justification For Why Not Higher Score:**

The revisions in the paper were substantial, to such an extent that it is not reasonable to expect an adequate judgement by reviewers during the short rebuttal period for this conference. The confidence of the raised scores was also fairly low. Note that the revisions cover previously raised issues on clarity on the workings of the proposed method, changes in proofs, and also significant changes in the results of the speed of the proposed method.

**Justification For Why Not Lower Score:**

N/A

---

### Decision · Program_Chairs · 2024-01-16

Reject